# Phosphorus Feast and Famine in Cyanobacteria: Is Luxury Uptake of the Nutrient Just a Consequence of Acclimation to Its Shortage?

**DOI:** 10.3390/cells9091933

**Published:** 2020-08-21

**Authors:** Alexei Solovchenko, Olga Gorelova, Olga Karpova, Irina Selyakh, Larisa Semenova, Olga Chivkunova, Olga Baulina, Elizaveta Vinogradova, Tatiana Pugacheva, Pavel Scherbakov, Svetlana Vasilieva, Alexandr Lukyanov, Elena Lobakova

**Affiliations:** 1Faculty of Biology, Lomonosov Moscow State University, 119234 Moscow, Russia; ogo439@mail.ru (O.G.); karpova_ov@mail.bio.msu.ru (O.K.); i-savelyev@mail.ru (I.S.); semelar@mail.ru (L.S.); olga.chivkunova@mail.ru (O.C.); baulina@inbox.ru (O.B.); lizavin@yandex.ru (E.V.); ismagulova@mail.bio.msu.ru (T.P.); cyano@mail.ru (P.S.); vankat2009@mail.ru (S.V.); loockart@mail.ru (A.L.); 2Ecology Research Laboratory, Pskov State University, 180000 Pskov, Russia; 3Institute of Natural Sciences, Derzhavin Tambov State University, 392000 Tambov, Russia

**Keywords:** luxury uptake, cyanobacteria, polyphosphate, transcriptomics, ultrastructure rearrangements, phosphorus starvation

## Abstract

To cope with fluctuating phosphorus (P) availability, cyanobacteria developed diverse acclimations, including luxury P uptake (LPU)—taking up P in excess of the current metabolic demand. LPU is underexplored, despite its importance for nutrient-driven rearrangements in aquatic ecosystems. We studied the LPU after the refeeding of P-deprived cyanobacterium *Nostoc* sp. PCC 7118 with inorganic phosphate (P*_i_*), including the kinetics of P*_i_* uptake, turnover of polyphosphate, cell ultrastructure, and gene expression. The P-deprived cells deployed acclimations to P shortage (reduction of photosynthetic apparatus and mobilization of cell P reserves). The P-starved cells capable of LPU exhibited a biphasic kinetic of the P*_i_* uptake and polyphosphate formation. The first (fast) phase (1–2 h after P*_i_* refeeding) occurred independently of light and temperature. It was accompanied by a transient accumulation of polyphosphate, still upregulated genes encoding high-affinity P*_i_* transporters, and an ATP-dependent polyphosphate kinase. During the second (slow) phase, recovery from P starvation was accompanied by the downregulation of these genes. Our study revealed no specific acclimation to ample P conditions in *Nostoc* sp. PCC 7118. We conclude that the observed LPU phenomenon does not likely result from the activation of a mechanism specific for ample P conditions. On the contrary, it stems from slow disengagement of the low-P responses after the abrupt transition from low-P to ample P conditions.

## 1. Introduction

Phosphorus (P) is an important nutrient central to storing and the exchange of energy and information in the cell [1,2]. At the same time, the availability of P in many habitats dwelled by cyanobacteria is scarce and/or fluctuating. Cyanobacteria developed a broad array of acclimations to cope with P shortage. One of the most widespread is the capability of taking up P in large excess of the current metabolic demand, termed “luxury P uptake” (LPU) [3]. The ability of the P-deprived culture to accumulate excessive amounts of inorganic polyphosphate (PolyP) after refeeding with P*_i_* is known as “hyper-compensation” or “phosphate overplus” [4,5].

The phenomenology and mechanisms of LPU in oxygenic phototrophs is dramatically underexplored in comparison with their responses to P starvation [6]. A few seminal studies have been carried out more than 50 years ago using eukaryotic microalgae like *Chlorella vulgaris* as a model [7,8]. The works on LPU in cyanobacteria were even more scarce, although remarkable exceptions exist [9]. It was established that P deprivation enhances the LPU capacity of the cyanobacterial cells during their subsequent refeeding with P [10,11]. The LPU capacity was also shown to be inversely related with filling up the cell internal capacity (or “quota”) for P [12].

Most of the inorganic phosphate (P*_i_*) taken up via LPU is stored in the cell in the form of osmotically inert polyphosphate (PolyP) granules [13], accommodating P sufficient for up to 20 cell divisions [14] and playing a plethora of other biological roles [1,4,15], including a potential role of energy storage [16]. In general, PolyP is accumulated when the P*_i_* influx into the cell is larger than its current metabolic demand and there is enough surplus energy (ATP). Overaccumulation of PolyP after refeeding of the P-deprived cells was documented in *Plectonema boryanum* [17,18], *Anacystis nidulans* [9], *Microcystis aeruginosa* [19], and *Calothrix parietina* [20]. The ATP is needed for pumping of the P*_i_* into the cell, unless the external P*_i_* concentration is high [21,22], and for the conversion of P*_i_* into PolyP [6,13]. The amplest source of energy for LPU is photosynthesis; dark respiration, or even fermentation, can also drive LPU and PolyP formation but with a lower efficiency [7].

Well-documented molecular mechanisms underlying the P-shortage responses include the upregulation of high-affinity transporters efficiently pumping P*_i_* into the cell, deploying extracellular phosphatases mobilizing organic P, and engaging intracellular mechanisms for the salvaging and sparing of P from, e.g., rRNA and/or phospholipids [2]. A bright example is the upregulation of the genes coding for extracellular phosphatases and P*_i_* transporters encoded by the genes from PHO regulon [23], as well as the gene cluster of phosphonate uptake and assimilation [24].

By contrast, the literature available to us at the time of this writing lacked descriptions of the specific mechanisms upregulated in cyanobacteria under ample P conditions. This gap in our knowledge contradicts with the importance of the cyanobacterial ability to absorb and accumulate P for nutrient-driven rearrangements in aquatic ecosystems such as the blooms of toxic cyanobacteria [25,26]. Furthermore, the capability of LPU can be exploited for the biotreatment of wastewater coupled with the production of environment-friendly biofertilizers [27,28]. To bridge this gap, we studied LPU in a nondiazotroph cyanobacterium *Nostoc* sp. PCC 7118 (which is essentially identical to a model strain PCC 7120 in the genomic sequence, excepting certain genes in control of heterocyst maturation [29,30]). We followed the ultrastructural rearrangements of the cell, along with the changes in gene expression patterns in transit from P starvation to ample P conditions, focusing on P uptake systems and the turnover of intracellular reserves of P, as well as on photosynthetic apparatus acclimations. We conclude that the LPU phenomenon in the P refeeding experiments is unlikely to be a result of the activation of the mechanisms specific for ample P conditions. It rather originates from the relatively slow deactivation of low-P responses after an abrupt transition from low-P to ample P conditions. The implications of this finding for biotechnology are discussed.

## 2. Materials and Methods

### 2.1. Microalgal Strains and Cultivation Conditions

In this work, we used the strain *Nostoc* sp. PCC 7118, a heterocyst-less mutant [29,31] of the model filamentous diazotrophic cyanobacterium *Nostoc* sp. PCC 7120, whose genome was published [30,32,33]. The precultures of *Nostoc* sp. PCC 7118 were grown in 0.75-L Erlenmeyer flasks containing 300 mL of BG-11 medium [34] at 25 °C and 40 μmol PAR photons m^–2^ s^−1^ in an Innova 44R (Eppendorf-New Brunswick, Framingham, MA, USA) incubator shaker. The precultures were maintained at the exponential phase by daily dilution with the same medium. The precultures were then scaled up in 0.6-L glass columns (4-cm internal diameters) under continuous illumination by LED lamps (4700 K) at 120 μmol PAR quanta m^–2^ s^−1^, as measured in the center of an empty column with a LI-850 quantum sensor (LiCor, Lincoln, NE, USA). The precultures were sparged with a CO_2_:air (1:50, *v/v*) mixture at a rate of 0.5 L min^−1^.

To obtain cultures with depleted internal P reserves, the preculture cells were harvested by centrifugation (1500× *g* for 7 min), washed twice with fresh P-free BG-11 medium (designated as BG-11–P), and resuspended in the same medium with the CO_2_ dosing valve.

The culture growth was monitored via the cell density, which was calculated using a hemocytometer. Average specific growth rate was calculated as follows:(1)$$\mu =\frac{\log\left(N_{1}\right)-\log\left(N_{0}\right)}{t_{1}-t_{0}}$$
where *N*_1_ and *N*_0_ are cell densities of the culture at times *t*_1_ and *t*_0_, respectively.

Cell dry weight (DW) was determined gravimetrically [35]. Contents of chlorophyll and total carotenoids (Car) were assayed spectrophotometrically, as described earlier [36]. Relative changes in the phycobilin amount were estimated as the ratio of OD at 620–624 nm (governed by phycobilins and Chl) to that at 678 nm (governed by chlorophyll) [37].

To obtain cultures with depleted internal P reserves, the cells were pelleted by centrifugation (1500× *g* for 7 min), washed with the BG-11–P medium, resuspended in the same medium, and incubated under the conditions described above in the P-free medium. The onset of P starvation was detected by a decline in the cell division rate (Figure 1). The cultures that showed no significant increase in cell number for at least three consecutive days under our experimental conditions were considered as P-depleted and used for further experiments. The refeeding experiments were initialed by an addition of KH_2_PO_4_ solution in distilled water to the culture to a final concentration of 400 µmol L^−1^. In certain experiments, several concentrations from the range 2300–100 µmol L^−1^ were used, as specified in the corresponding figure legends.

### 2.2. The Studies of the Fast P_i_ Uptake Phase

The cells with depleted P reserves obtained as described above were incubated for 1 h under illuminations (250 μmol PAR quanta m^–2^ s^−1^) or in darkness at 25 °C or 4 °C. After the acclimation, the concentrated KH_2_PO_4_ solution was added to the final concentration of 1000 µmol L^−1^ of P*_i_*. The first suspension sample was taken as soon as possible after the P*_i_* refeeding and filtered through a syringe-driven filter with a PTFE membrane (Merck-Millipore, Burlington, MA, USA). The actual sampling intervals comprised 18, 36, 60, 120, 180, 300, 420, 600, 780, and 960 s. The resulting filtrate was immediately frozen in liquid nitrogen and stored at −20 °C before assaying of the residual P*_i_* concentration in the medium by ion-exchange HPLC (see below).

### 2.3. Spectral Measurements

Absorbance spectra of the cyanobacterial cell suspensions were recorded with a Cary 300 Bio spectrophotometer (Agilent, Santa Clara, CA, USA) equipped with an integrating sphere. The measured spectra were scattering-corrected using the method by Merzlyak et al. [38].

### 2.4. Electron Microscopy and Cell Morphometry

The microalgal samples for transmission electron microscopy (TEM), including nanoscale elemental analysis in TEM using energy-dispersive X-ray (EDX) spectroscopy and energy-filtered (EF)TEM cell section mapping, were prepared and processed as described previously [39]. All quantitative morphometric analyses were done as described previously [40]. Briefly, at least 3 samples from each treatment were examined on sections through the cell equator or subequator. The subcellular structures and inclusions were counted on the sections. Linear sizes, as well as the subcellular structure area, were measured on the TEM micrographs of the cell ultrathin sections (*n* = 20) using Fiji (ImageJ) v. 20200708-1553 software (NIH, Bethesda, MA, USA). In the quantification of PolyP, only granules >25 nm in diameter were counted.

### 2.5. Assay of Phosphate in the Medium: Total Phosphorus and Polyphosphate in the Cells

The routine assessments of nitrate (to check that the culture had an adequate supply of nitrogen throughout the experiment) and orthophosphate ion concentrations were done using Thermo Dionex ICS 1600 HPLC (Thermo-Fisher, Sunnyvale, CA, USA) fitted with a conductivity detector and IonPac AS12A (5 µm; 2 × 250 mm) anionic analytical column with an AG12A guard column (5 µm; 2 × 50 mm), according to an earlier published method [41]. Briefly, the column temperature was maintained at 30 °C. The ions were eluted isocratically with a 2.7-mmol L^−1^ sodium carbonate/0.3-mmol L^−1^ sodium bicarbonate buffer (flow rate of 0.3 mL min^−1^). The residual P*_i_* and nitrate contents in the medium were additionally verified each third day with standard cuvette tests LCK 380 and LCK 350 (Hach Lange, Dusseldorf, Germany). At certain time points of the experiment, the total cell P content was chemically assayed using the molybdenum blue chromogenic test and sample preparation procedures described in [42,43]. It was found that the accumulation of P in the cells corresponded, with a reasonable precision (ca. 10%), to the depletion of P*_i_* in the medium under our experimental conditions. Since the analysis of the residual P*_i_* is simpler, it was routinely employed in this work. The intracellular PolyP content was assayed by 4′,6-diamidino-2-phenylindole (DAPI) staining ([39]; for more detail, please refer to online Appendix A).

### 2.6. Sequencing of Nostoc sp. PCC 7118 Genome

To verify the identity of the genes of interest in the strains *Nostoc* sp. PCC7118 and *Nostoc* PCC 7120 (whose genome was used here as the reference), we obtained the short reads of the genomic DNA of PCC 7118 and mapped it to the published genome of PCC 7120 [33] using Geneious Prime software (Biomatters, Auckland, New Zealand).

Towards this end, genomic DNA was isolated from the aliquots of the exponentially grown P-sufficient precultures of PCC 7118 (see above). Genomic DNA was extracted from 100-mg cell samples (wet weight) with a GeneJET Genomic DNA Purification kit (Thermo Scientific, Waltham, MA, USA) according to the manufacturer’s protocol. Prior to isolation, cells were resuspended in the lysis buffer and disrupted in the FastPrep-24 5G grinder (MP BioMedicals, Irvine, CA, USA). The DNA sample quality was evaluated by electrophoresis in 1% agarose gel with ethidium bromide staining.

Genomic DNA was fragmented by sonication. Illumina libraries for whole-genome sequencing were constructed using a NEBNext DNA Library Prep Master Mix Set for Illumina (New England Biolabs, Ipswich, MA, USA) according to the manufacturer’s protocol. Whole-genome shotgun libraries were sequenced on Illumina HiSeq 2500 (paired-end 150 nt). The Illumina sequences reported in this paper have been deposited in the National Center for Biotechnology Information’s Sequence Read Archive (accession no. PRJNA626624).

### 2.7. Studies of Gene Expression

Sequencing of the whole transcriptome of *Nostoc* sp. PCC 7118 for discovering and selecting the genes potentially relevant to the phenomena observed during the luxury uptake of P was carried out as described earlier [44]. For the transcriptome analyses, cells were harvested from (i) the P-sufficient exponentially growing precultures, (ii) P-sufficient early stationary precultures, (iii) P-starved cultures just before the P*_i_* refeeding, (iv) one day after refeeding (recovery and exponential cell division), and (v) seven days after refeeding (the onset of the stationary phase due to cell self-shading). Total RNA was extracted from the cell samples [45], treated with an Ambion Turbo DNA-free kit (Thermo Fisher, Waltham, MA, USA), and quality-checked with an Agilent Bioanalyzer (Agilent, Santa Clara, CA, USA). Ribosomal RNA was removed from total RNA (0.2 mg) using a Ribo-Zero rRNA Removal Kit for Gram-negative Bacteria (Epicentre, Illumina, San Diego, CA, USA).

Sequencing was performed with an Illumina HiSeq 2000 System, and approximately 10 million 100-bp paired-end reads per replicate sample were mapped to the reference genome *Nostoc* sp. PCC 7120 [33]. The relative abundance of transcripts has been calculated using Geneious (Biomatters, Auckland, New Zealand) software with default parameters. The DeSeq2 algorithm, which compares FPKM (Fragments per Kilobase of exon per Million fragments mapped) values between treatments, allowed fold changes (FC) in the expression for each gene and the statistical significance (cutoff: *p* > 0.05) of these changes to be assessed [46]. The changes have been calculated as the log_2_FC of FPKM relative to the P-sufficient exponentially growing preculture (see above).

Differentially expressed genes of interest were identified based upon a keyword search in CyanoBase (bacteria.kazusa.or.jp/cyanobase/) and verified against the reference genome annotation (PCC 7120). The Illumina sequences reported in this paper have been deposited in the National Center for Biotechnology Information’s Sequence Read Archive (accession no. PRJNA626624).

The presence and the differential expression levels of the selected genes were verified by a real-time polymerase chain reaction (qRT-PCR; for primers, see Appendix A). The qRT-PCR was performed using the QuantiTect SYBR Green PCR Kit (Qiagen, Hilden, Germany) according to the manufacturer’s recommendations, the QuantStudio 7 Flex Real-time PCR System (Thermo Fisher Scientific, Waltham, MA, USA), and the Applied Biosystems QuantStudio^™^ Real-time PCR Software Version 1.3 (Thermo Fisher Scientific, USA). All measurements were carried out with two biological and two analytical replicates. The expression of the target genes at different stages after refeeding the cells with P*_i_* was calculated relative to that recorded in the cells of the exponentially growing P-sufficient preculture. The obtained data were processed using the Thermo Fisher Cloud Data Analysis software (Thermo Fisher Scientific, USA) with the default parameters.

### 2.8. Statistical Treatment

All experiments were carried out in three biological replicates, with two analytical replicates for each of them. In the figures, average values, together with standard deviations, are presented, unless otherwise stated. The significance of differences was tested using ANOVA from the analysis tool pack of the Excel (Microsoft, Redmond, WA, USA) spreadsheet software.

## 3. Results

### 3.1. Changes in the Growth Rate of and Light Absorption by the Culture during P Starvation and Recovery from It

The P deprivation exerted no measurable effect on the rate of cell division during the first two to three d of the experiment (Figure 1, left scale; average growth rate, µ = 0.35), suggesting the presence of considerable P reserves in the cells of the preculture (the cell P content of the preculture comprised 2.1% of the cell DW). To avoid a slow-down of the growth due to self-shading of the cells, the cultures were diluted with the BG-11–P medium to maintain the OD_678_ below 1.0. After the dilution, the cell division rate declined and eventually stopped manifesting the depletion of the internal P reserves (to 0.9% of cell DW) and the onset of P starvation. Within three days after the cessation of cell division and DW accumulation, the cultures were refed with P*_i_* in the form of KH_2_PO4 (Figure 1, right scale). After a short (4–6 h) lag phase, the cell division resumed; the average growth rate during this period comprised 0.46. Within the first 24 h after P*_i_* refeeding, the culture consumed, on an average, 18 pmol P*_i_* cell^−1^. During the next six days after refeeding, the uptake rate slowed down, comprising 3.7 pmol P*_i_* cell^−1^. By the seventh day after the refeeding, the rate of cell division (µ = 0.05) and accumulation of dry weight (curve *2* in Figure 1) started to decline, manifesting the onset of the early stationary phase due to light limitations.

In-line with our observations on the growth rate, P deprivation had little effect on the culture growth and its absorbance spectra in the beginning of the experiment (Figure 2A). A steady increase in light absorption in the blue region of the spectrum (see also insert in Figure 2A), suggesting an increase in carotenoids relative to chlorophyll (insert in Figure 2A), took place during P starvation. A pronounced decline of the absorbance in the band centered at 620 nm governed by phycobilins was also documented. At more advanced stages of P starvation, a pronounced decline in chlorophylls on the background of retention of carotenoids took place, evident as a decline in A_678_ and increase in the ratio A_480_ A_678_^−1^, respectively (Figure 2A). Refeeding with P*_i_* reversed the changes observed during P starvation (Figure 2B). Thus, the cells reaccumulated chlorophylls and phycobilins (insert in Figure 2B), whereas the contribution of Car to the light absorption of the cell suspensions declined in comparison with that of other pigments to the level typical of the P-sufficient cultures.

### 3.2. The Kinetics of P_i_ Uptake and PolyP Formation during after Refeeding of the P-Starved Cultures

After the refeeding of the P-starved cells with P*_i_*, the nutrient was rapidly absorbed, showing a typical biphasic kinetic of uptake [13] (Figure 1, Figure 2, Figure 3 and Figure 4). We studied the P*_i_* uptake kinetics as a function of the concentration of P*_i_* added, light, and temperature. After refeeding, concentration of P*_i_* in the medium oscillated during the first 60 min (Figure 3). The initial decline occurring as a result of very fast uptake was followed by a transient increase in the external P*_i_* concentration and, later, by a slower uptake phase. The higher the P*_i_* concentration added during the refeeding, the higher the magnitude of these oscillations (Figure 3). Judging from the residual P*_i_* concentration in the medium, a large amount of P (15–60% of the added P*_i_*, depending on the external P*_i_* concentration) enters the cells during the first hour after refeeding (approx. 4% of the cell DW, which comprised 0.49 g L^−1^; Figure 3).

To better understand the nature of the observed kinetics, we followed the first-phase P*_i_* uptake (the rapid decline) under physiological conditions as well in darkness and/or at 4 °C. Notably, the fast phase of P*_i_* uptake by the P-starved cells of the cyanobacteria took place independently of the light availability, even under the chilling temperature (Figure 4).

After the P*_i_* refeeding, the cells restored their PolyP reserves. The kinetics of the PolyP formation displayed two maxima. A transient increase in PolyP formation was detected approximately 4–6 h after the P*_i_* refeeding (Figure 5). This period corresponded to the lag phase when the culture was in-transit from P starvation to recover from it and cell divisions did not yet resume (Figure 1). Later, when the cyanobacterial cells stared to divide, the PolyP level in the cells declined, although P*_i_* was still available in the medium. The second increase in the PolyP cell content took place upon the onset of the stationary phase when the cell division rate slowed down again (Figure 5). The data on the depletion and subsequent biphasic accumulation of PolyP were confirmed by analytical TEM (see, e.g., Appendix A).

### 3.3. Ultrastructural Hallmarks of P Starvation, Recovery, and Luxury Uptake of This Nutrient

The P-sufficient cells of *Nostoc* sp. 7118 precultures grown in the P-sufficient BG-11 medium displayed a cell organization typical for cyanobacteria (Figure 6A,B; see also [47]. The cell wall consisted of the outer membrane and peptidoglycan layer. Most of the cytoplasm volume was occupied by paired thylakoid membrane groups (consisting of 4–6 thylakoids) surrounding the areas of the nucleoid with compact DNA filaments and numerous ribosomes. The phycobilisomes featured a low contrast due to the relatively high electron density of the cytoplasm surrounding them.

The nucleoid areas harbored polyhedral carboxysomes, which contain RuBisCO. The reserve structures were represented by regularly distributed moderately ample α-granules (glycogen), β-granules (lipid droplets), and polyhydroxybutyrate (PHB) granules. On the periphery of the PHB granules close to the nucleoid, and in the interthylakoid space, the electron-dense granules were located, identified by their EDX spectra as P-containing inclusions (PolyP) [48]. The granules of cyanophycin were small (0.015 ± 0.001 μm^2^ on an average) and encountered on less than 60% of the studied cell sections (Figure 6).

The P deprivation triggered a progressive reorganization of the protoplast but not of the cell wall (Figure 6C–G). The overall electron density of the cell sections declined, apparently due to a decline in the amount of soluble proteins and the number of small electron-dense structures. In particular, the nucleoid in the P-starved cells became sparse, and the number of ribosomes decreased. A pronounced reduction of the photosynthetic apparatus was documented. The thylakoids became shorter and/or fragmented. The phycobilisomes became more distinct due to a decline in the electron density of the surrounding cytoplasm, but they were smaller than those in the cells of P-sufficient cultures (base width × height of 43 ± 2 nm × 29 ± 1 nm; see below). The reduction of the photosynthetic apparatus also brought about a three-fold decline in the carboxysome abundance and total area (Figure 6). 

The granules of glycogen and PHB disappeared; β-granule abundance did not change considerably (notably, they were localized in close contact with the decomposing thylakoid membranes (Figure 6F,G)). At the same time, large amounts of cyanophycin granules were recorded in 100% of the studied cell sections (Figure 6C,D and Figure 7).

After the replenishment of P*_i_* in the culture medium, ultrastructural changes were recorded as early as 4 h after the refeeding (Figure 7A–C). Those were essentially comprised by the reversal of the changes observed during P starvation (Figure 7). Namely, the electron density of cytosol increased, apparently due to the increase of the soluble protein content, the nucleoid became more compact, and the number of ribosomes increased. The ultrastructure of thylakoids and phycobilisomes returned to that typical of the P-sufficient cells, suggesting the recovery of the photosynthetic apparatus, including the light-harvesting antenna. Thus, the phycobilisome size increased (to the base width × height of 54 ± 2 nm × 36 ± 1 nm) upon 4 h after P*_i_* refeeding, but complete recovery of the thylakoid membrane system and carboxysomes was achieved by the seventh day after refeeding (Figure 7D,E).

Glycogen granules reappeared, suggesting that not only the structure of the photosynthetic apparatus but, also, its function has recovered. The cyanophycin granules did not change significantly during the first hours after refeeding but declined 11-fold upon the resumption of cell division; still, their area was four times than in the cells from the preculture (Figure 8). 

Remarkably, numerous electron-dense granules comprising P (likely in the form of PolyP) were formed between the thylakoids within the nucleoid zone and in the PHB granules during the fast phase of the luxury P*_i_* uptake (first 4 h) by the prestarved cells of *Nostoc* sp. PCC 7118 (Figure 7 and Appendix A). The amount of the PolyP granules >25 nm in diameter was 11 times higher (total area was 3.8 times higher) in the prestarved and refed cells than in the cells from the P-sufficient precultures (Figure 8). Interestingly, the amount of the PHB granules was low in the P-deprived cells, but in the refed cells, it was 3.5 times higher than in the cells from P-sufficient precultures (Figure 8). 

Collectively, the features of the ultrastructure aligned well with the events during P starvation and recovery after refeeding of the P-starved cells with P*_i_*. The cell accumulated a large amount of P-containing inclusions shortly after P*_i_*-refeeding, when active cell division did not yet resume. The recovery of the cell ultrastructure to that observed in the P-sufficient preculture was largely completed by the seventh d after the P*_i_* refeeding (Figure 6A,B and Figure 7D,E). However, the reserve inclusions (PHB, PolyP, and cyanophycin granules) remained at the levels higher than those in the preculture (Figure 8).

### 3.4. Dynamics of Gene Expression during P starvation and Recovery

#### 3.4.1. Overview of the Transcriptome Analysis

We verified that the PCC 7118 clone used in this work is indeed close (at least regarding the genes involved in this study) to PCC 7120 by mapping the short genomic reads obtained for in our experiments to the reference genome of PCC 7120. As a result, a good (>20, on an average) coverage of the genes of interest was achieved, enabling us to assume these regions of the PCC 7118 genome to be identical to those of the reference genome. Based on this assumption, the functional annotation of the reference genome was tentatively accepted for the corresponding genes of PCC 7118. 

The results of RNASeq for selected genes of interest agreed with those of qRT-PCR (Appendix A). More than 45 million (95%) of the obtained short reads were mapped to the annotated reference genome. The cutoff value of ≤ 0.05 for the false discovery rate and four (log_2_FC = 2) as the fold change of the expression level were generally applied (for the regulatory genes of interest, the cutoff FC value was not applied). As a result, a total of 376 genes for the P-sufficient early stationary preculture, 358 for the P-starved culture (−P), and 436 and 330 genes for the cultures recovered from P starvation for one day (log + P) or seven days (stat + P) after P*_i_* refeeding were differentially expressed as compared with the cells of the preculture. The genes of interest with statistically significant differentially expressed transcripts were divided into categories according to their functional roles, as considered below (Appendix A). Notably, a significant part of the genes with |log_2_FC| > 2 belonged to the genes with unknown functions and/or coding for hypothetical proteins (not shown).

#### 3.4.2. Phosphorus Uptake and Intracellular Storage

The genome of *Nostoc* sp. 7118 harbors two-component regulatory systems, including the homologs of the genes *phoR*, *phoS*, and *phoU* responsible for the regulation of P transport and acquisition [49] and two pstABCS operons and genes coding for the phosphonate transporters and phosphatases ([23]; Table 1; see also Appendix A). High-affinity P*_i_* transporter systems pstABCS enable P uptake over a wide range of concentrations; they are similar to those in heterotroph bacteria, i.e., *Escherichia coli* [50] and were discovered in many cyanobacterial species. The genes coding for the proteins related to the P acquisition and assimilation from alternative P sources, such as organic phosphate, phosphonate, and phosphites, were predicted and/or characterized in the genomes of several cyanobacterial species [2,23,51], although their functioning has not been elucidated in full detail. Under our experimental conditions, both pstABCS operons were upregulated during P starvation and rapidly repressed after P*_i_* refeeding, although their residual expression level remained significantly higher than in the P-sufficient preculture (Table 1). A little variation of the expression of the well-known regulatory elements homologous to *sphS* (all4502), *sphR* (all4503), and *phoU* (all4501) was revealed, regardless of the experimental conditions employed (Appendix A). This observation is compatible with the current understanding that the regulatory function of the corresponding proteins is implemented via their phosphorylation-dephosphorylation [49].

The genome sequence of *Nostoc* sp. PCC 7118 (PCC 7120) indicated the capability of this strain of using P from alternative sources, including organic P and phosphonates. Although P*_i_* was the only P source for the culture, the ortholog of the extracellular phosphatase all2843 and alr2234 (a homolog of *phoD*) was almost 30-fold upregulated in the P-starved cells in comparison with the P-sufficient preculture, representing one of the most upregulated genes under P starvation conditions. At the same time variation of the alkaline phosphatase alr1686 (a *phoA* homolog), the expression was insignificant (Table 2), whereas the external nuclease *nucH* was upregulated. Three of the genes in control of phosphonate transport, *phnC* (all2230), *phnD* (all2228), and *phnE* (alr2226), were upregulated under phosphorus starvation; gene *phnG* was downregulated at the exponential growth following the P*_i_* refeeding (Appendix A).

The expression profile of the genes coding for PolyP kinase alr3593 (*ppk1*) was somewhat different from the observed kinetics of PolyP accumulation. It possessed a single maximum (Appendix A and Table 2) roughly coinciding with the transient peak of PolyP accumulation recorded during the fast phase of LPU (Figure 5), but there was no upregulation of these genes overlapping with the second peak of PolyP accumulation detected at the stationary phase (Figure 5). A similar expression pattern was revealed for alr2191 (*ppk2*) by real-time (RT)-qPCR (Appendix A) but not with RNASeq (Table 2). The genes coding for the enzymes of the cyanophycin turnover showed contrasting patterns of expression; at least one of them (*cphA1*) was upregulated in the P-starved cells (Appendix A).

#### 3.4.3. Photosynthetic Apparatus and Central Metabolism

The photosynthetic apparatus of *Nostoc* sp. PCC 7118 consists of photosystems I and II, cytochrome b_6_/*f*, intersystem electron carriers, and F-type ATPases; the photosystems include phycobilisomes—the light-harvesting complexes containing allophycocyanins and phycocyanins [53]. In our study, the expression of genes of the photosynthetic apparatus was dramatically reduced during P starvation. The most downregulated ones were the genes coding for the phycobilin antenna components (Appendix A). The genes coding for the small and large subunits of RuBisCO and its activase (Appendix A) were also downregulated, manifesting a decline in the light absorption and photosynthetic carbon capture (see also Figure 2A). This trend was rapidly reversed after the P*_i_* refeeding in agreement with the observed recovery of the photosynthetic apparatus (Figure 2B).

In addition, the transcription of many genes participating in core metabolism—e.g., those coding for phosphoenolpyruvate carboxylase or glucose 6-phosphate dehydrogenase—declined during P starvation. However, the expression of those genes was upregulated within one day after refeeding, i.e., when the cell divisions were resumed, suggesting the acceleration of the cell metabolism after quiescence during P starvation (not shown). Genes of the ribosomal proteins displayed similar expression patterns in-line with the recorded kinetics of the culture growth and previously described responses to various nutrient stresses [54,55].

## 4. Discussion

Many photosynthetic microorganisms, including cyanobacteria, evolved in nutrient-poor environments or environments with varying nutrient concentrations. As a result, they acquired a broad array of acclimations to scarce and/or fluctuating nutrient availability, including the capability for luxury uptake. Although this phenomenon has been known for quite a long time [3], many aspects of LPU remain largely unknown to date. Furthermore, phosphorus eutrophication is often named among the root causes of harmful cyanobacteria blooms [56,57].

Early ultrastructural descriptions of the PolyP formation during LPU were obtained in cyanobacteria [10]. Here, we attempted to obtain a deeper insight into LPU through the analysis of changes in the cell ultrastructure and gene expression displayed by the cyanobacterium *Nostoc* sp. PCC 7118 in transit from P starvation to ample P conditions. We also complemented the traditional transmission electron microscopy (TEM) with analytical TEM, providing reliable information on the subcellular distribution of P-rich inclusions [48], including polyphosphate formed after the refeeding of P-starving cells [58]. We leveraged these data to search for specific acclimations to ample P conditions. The alternative hypothesis was that the observed LPU phenomenon results from a slow disengaging of the acclimations to P shortage after an abrupt increase of the P*_i_* availability in the medium.

The analysis of the available literature on LPU in cyanobacteria and eukaryotic microalgae (e.g., [6,7,9,59]) reveals a lot of inconsistencies pointed out, e.g., in [21]. Still, it shows that the observed characteristics of LPU strongly depend on the cultivation conditions and P nutrition prehistory of the culture. Thus, cyanobacterial cultures subjected to P depletion, even for a short time, acquire the capability of the very rapid uptake of P*_i_* [22]. At the same time, the quantitative and kinetic parameters of P uptake seem to be related to the available total capacity (or quota; see, e.g., [60]) of the cell for this nutrient. The cell P quota is a compound characteristic comprised of several pools, e.g., DNA, RNA, phosphometabolites, and PolyP. These pools differ dramatically by their size and flexibility of response to P availability in the medium [9], posing challenges to the reliable estimation of the actual cell P quota size. Therefore, to make our experiments on LUP as deterministic as possible, we used the cells with the P quota depleted by depriving them of P until the cell division is ceased [59]. This approach also allows us to reconstruct the events taking place in natural cyanobacterial populations upon the abrupt increase of P availability, e.g., as a result of upwelling [26].

Accordingly, the first part of the experiment was comprised of starving the *Nostoc* sp. PCC 7118 culture of P until the cell density stopped increasing. This stage was completed within eight to nine d (Figure 1). In-line with previous reports on cyanobacteria such as *Anabaena*, *Synechococcus*, *Microcystis*, and *Prochlorococcus* [44,53,54,61,62], P starvation promoted the onset of the stationary phase (Figure 1). As noted above, the specific response to P starvation was constituted by the upregulation of the P acquisition systems, including those in control of the P*_i_* uptake that are well-studied both in model heterotroph bacteria [50] and cyanobacteria [23,49]. The cyanobacterial cells, including those of the strain studied in this work, take up P*_i_* through the *pstABCS* transporter family, including ATP-driven ABC-type P*_i_* pumps (Table 1, [22]) crucial for P accumulation [63]. It is regulated by the two-component system operating via the phosphorylation/dephosphorylation of *phoB* and *phoR* or their orthologs *sphS* and *sphR* [49], with participation of the repressor *phoU* (or *sphU*). The same regulatory system also orchestrates the acquisition of P from organic sources [23,49]. 

Interestingly, in the P-starving culture of *Nostoc* sp. PCC 7118, the expression of both *pstABCS* gene clusters present in the genome was at approximately the same level, whereas, in other cyanobacteria, one of the operons did not respond to P shortage [62,64]. A dramatic response to P-deprivation with subsequent P*_i_*-refeeding was recorded in some (but not all) alkaline exophosphatases (Table 2), which was in-line with previous reports [62]. The capacity of the cyanobacterial cells to acquire P under P limitations is enhanced by the excretion of exophosphatases, like those encoded by the all2234, all2843, or alr4976 genes (Table 2, [65]). The expression pattern of *phoD* and other phosphatase genes indicates that *phoD*, and likely *nucH*, are mainly responsible for the mobilization of external organic P, whereas other phosphatases play less important roles in this process, as was suggested by the results obtained for other cyanobacteria [24,54]. Some of the genes responsible for the utilization of P in form of phosphonates (Table 2, [66]) were upregulated in P-starving *Nostoc* sp. PCC 7118 and repressed upon P*_i_*-refeeding. In contrast to the strain studied in this work, in other cyanobacteria, e.g., in *Anabaena* 90 [62], the *phn* cluster was downregulated, suggesting that the *phn*-genes are not under the control of the *pho*-regulatory system.

In parallel with the specific responses to P-shortage, general stress responses can be observed in cyanobacteria under nutrient depletion [54,55]. Under nutrient stress, the nondiazotrophic cyanobacteria cells become metabolically quiescent, shutting down the activity of the central metabolism (glycolysis, pentose phosphate pathway, glycogenesis, and carbon fixation) and photosynthesis apparatus [54,67,68]. Notably, the cells of PCC 7118 seem to retain the nitrogen and/or carbon liberated from the proteins degraded during P starvation in the temporary storage comprised of the cyanophycin granules; a similar phenomenon was recorded in other P-deprived cyanobacteria (see, e.g., [10,11,58]). Notably, the culture coloration changes into yellow-green due to the loss of photosynthesis pigments [69], but changes on the molecular level are detectable even within a few hours [54,68]. This was also the case in our experiments (Figure 1 and Appendix A).

The recovery of the cell from P starvation and LPU accompanying this process was induced by the addition of P*_i_* to the P-starved culture. To a considerable extent, the phenotypic picture of the recovery was comprised of the inversion of the aforementioned changes recorded during P starvation. Thus, the photosynthetic apparatus recovered within a week after P*_i_*-refeeding, as did the studied parameters of the ultrastructural organization of the cell (Figure 1, Figure 6, and Figure 7); however, mobilization of the nutrient reserves stored in the cell inclusions somewhat lagged (Figure 8).

The kinetics of P*_i_* uptake during the recovery period displayed two distinct phases. The first phase was characterized by a rapid, partially reversible P*_i_* uptake (Figure 3); its duration was comparable with the time of filling up of the cell P*_i_* pool (a half-time of ca. 4 min [21]). The influx of P*_i_* at this phase depended on the external P*_i_* concentration; this observation is in-line with previous findings in *Synechococcus* sp. strain R2 [21,22]. The effects of the added P*_i_* concentration, light, and temperature recorded in this study suggest that this phase of luxury P*_i_* uptake reflects the operation of a passive transport requiring no energy input in the form of ATP. One can speculate that this phase results from the large inflow of P*_i_* through the high-affinity transporters ample in the cytoplasmic membrane of cyanobacteria acclimated to a low P availability. This phase was not documented in previous studies due to an insufficient time resolution of the measurements. Importantly, one cannot rule out a certain contribution of a low-affinity P*i* transport to the LPU phenomenon. However, the data obtained do not support a sizeable contribution of these transporters, since the cells acclimated to ample P*_i_* did not display a massive P*_i_* uptake from the P*_i_*-replete medium (two to seven days after the P*_i_*-refeeding). Furthermore, the rapid P*_i_* uptake and transient PolyP accumulation were documented only in the cells with upregulated high-affinity P*_i_* transporters. Still, some constitutive uptake via low-affinity transporters can take place, since a buildup of PolyP occurred when cell division slowed down (six to seven d after the P*_i_*-refeeding). Moreover, there could be still unknown Pi transporters hiding in the uncharacterized part of the Nostoc genome.

The observed kinetics of P*_i_* uptake is well-described in terms of the model developed by Falkner and Falkner [13], presuming a biphasic kinetics of uptake when the concentration of the external P*_i_* was much higher than the thermodynamic threshold for P*_i_* transport into the cells (which was deliberately made the case in our refeeding experiments). 

Like other cyanobacteria, *Nostoc* sp. PCC 7118 has a cell envelope consisting of an outer membrane, a peptidoglycan layer, and a cytoplasmic membrane. The first step in the uptake of P*_i_* by cyanobacteria is comprised by its transport through the outer membrane. However, the mechanisms of this step (the type, representation, and specificity of corresponding porins resembling, e.g., *phoE* in *Escherichia coli*) remain obscure in *Nostoc* sp. PCC 7118. In addition, periplasmic transporter proteins can contribute to the bulk kinetics of P*_i_* uptake and its transport into the cell, e.g., by increasing the local P*i* concentration in the vicinity of the cytoplasmic membrane. Genome mining revealed the presence of potential periplasmic transporters and P*_i_*-binding proteins in PCC 7118. Some of them displayed small changes in their expression (alr1094; Appendix A), whereas the expression of other (see, e.g., all0917; Appendix A) changed considerably under the studied experimental conditions.

Obviously, these transporters can transfer a large amount of P*_i_* into the cell before they are downregulated. The large amount of P*_i_* rapidly entering the P-starved cyanobacterial cell during LPU can fatally displace the equilibria of important metabolic reactions. This risk is mitigated when a considerable part of the P*_i_* taken up by the cell during LPU is converted into PolyP. Although the cell reserves of PolyP were depleted after P starvation, there were cells with measurable levels of PolyP (Figure 5), which is typical for P-starving cyanobacterial cultures [70]. In such a situation, cyanobacteria rapidly convert the absorbed P*_i_* into PolyP, the main P reserve in the cell. This suggestion is compatible with the estimated time of P*_i_* turnover in the cyanobacterial cell (4–10 min, [21]). The rapid conversion of the absorbed P*_i_* is important (i) to avoid the disturbance of the metabolism by a large internal concentration of P*_i_* and (ii) to preserve the P for the next generation of cells. On the other hand, the excessive formation of short-chain PolyP must be avoided, since the latter can exert a toxic effect [71,72]. This speculation is supported by the transient increase in PolyP observed in this work and in other experiments with the refeeding of P-starved photosynthetic cells [59]. 

We hypothesize that the rapid conversion of the surplus P*_i_* into PolyP might be carried out with the participation of PolyP kinases, the key enzymes of PolyP metabolism [1]. Polyphosphate kinases are encoded by the genes of the *ppk* family playing an important role in the luxury uptake of P. Thus, the overexpression of transgenic *ppk* from a cyanobacterium *Microcystis aeruginosa* NIES-843 led to an enhanced P uptake by *Pseudomonas putida* KT2440 from wastewater in a sequencing batch biofilm reactor (SBBFR) [73]. The genome of the studied strain contains two genes encoding polyphosphate kinase enzymes *ppk1* and *ppk2* (Table 2 and Appendix A; see also [1]). Notably, in *Pseudomonas aeruginosa* PAOM5, the nucleoside diphosphate kinase activity of *ppk2* is 75 times higher than the PolyP elongation activity [74], so this enzyme is likely involved in the mobilization of PolyP in the cell. This is compatible with the observed expression pattern of *ppk2* in *Nostoc* sp. PCC 7118 under our experimental conditions. We observed only one maximum of the expression of the corresponding genes following the refeeding with P*_i_*; as in other P-starving cyanobacteria, e.g., studied in [75], the expression levels of these genes are closely correlated. Notably, there was a second maximum of PolyP accumulation documented in an early stationary P-sufficient culture (ca. seven d after P*_i_*-refeeding) not accompanied by a corresponding increase in PolyP kinase-encoding genes. It is possible to think that the conversion of P*_i_* to PolyP at this stage does not require a significant upregulation of *ppk* family genes.

It is believed that PolyP is mobilized via enzymatic hydrolysis by *ppx*-encoded exopolyphosphatase (PPX), with a subsequent hydrolysis of the liberated pyrophosphate by *ppa*-encoded inorganic pyrophosphatase (PPA). It is also possible that PolyP can be metabolized by the direct phosphorylation of sugars with the participation of a polyphosphate-dependent gluco(manno)kinase encoded by all1371 [76,77], showing a moderate upregulation (Table 2) during the exponential phase of growth after P*_i_*-refeeding.

A deep understanding of the LPU mechanisms and PolyP turnover is also important for the biotechnological application of cyanobacterial cultures. One of the emerging fields is the bio-capture of P from waste streams and its conversion into environmental friendly biofertilizer [27,78,79]. The results of the present study provide the informed selection of cultivation conditions to achieve the highest efficiency of P bio-removal and enrichment of the biomass with PolyP. The latter is important, since the P-rich biomass gradually decomposed by soil microbial phosphatases acts as a slow-release P fertilizer [80]. As was previously shown for eukaryotic microalgae, the P-starved cells best capable of P*_i_* uptake are potentially suitable for the post-treatment of wastewater [59]. 

Finally, one should realize that the insights generated by a single method, even so powerful as advanced omics techniques, are limited. Thus, only a limited correlation was found between the transcript and protein abundances revealed in the transcriptomic and proteomic experiments with a diazotrophic cyanobacterium [62]. Comprehensive approaches combining the strengths of molecular biology methods with those of structural biology (see, e.g., [81]) and conventional microbiology methods are better capable of resolving the “big picture” of complex biological phenomena like LPU.

## 5. Conclusions

Luxury phosphorus uptake is among the key acclimations of cyanobacteria to fluctuating P availability in nature. It turns out to be a complex process comprised of several distinct mechanisms. The increased P*_i_* uptake capability of the cyanobacterial cells acclimated to P shortage stems likely from the presence of high-affinity P*_i_* transporters, which continue to function for some time after an abrupt increase in external P*_i_* independently of light and/or temperature. The transient accumulation of PolyP observed shortly after P*_i_*-refeeding of the P-starved cells of *Nostoc* sp. PCC 7118 seems to result from an “emergency” upregulation of PolyP biosynthesis in response to the upsurge of P*_i_* in the cell. Upon the resumption of rapid cell division, these PolyP reserves are metabolized. The accumulation of PolyP starts again after the slow- down of cell division when P*_i_* is still ample in the medium.

Overall, the P shortage leads to a number of regulatory, functional, and structural changes to (i) acceleratethe P*_i_* uptake, (ii) mobilize external P resources, and (iii) spare intracellular P resources. These rearrangements render the cell primed for P hyperaccumulation manifesting itself as LPU upon resupplementation with P*_i_*. Still, this study did not reveal specific mechanisms induced in response to the elevated P availability; the observed manifestations of LPU are obviously the “extensions” of the known acclimations to P shortage (see, e.g., [82]). One can conclude that the observed global reprogramming of gene transcription and, hence, cell metabolic rearrangement is what constitutes the “adaptive responses of phosphate-deficient cells to an abrupt rise of the external phosphate concentration [comprising the] intracellular self-organization process, in which the kinetic and energetic properties of the phosphate uptake system are altered in a complex manner that reflects intracellular information processing about alterations of [the] phosphate supply” described in the seminal works of Falkner et al. [13,14]. On the other hand, the key to the specific acclimations to ample P conditions still might remain hidden in the genome of cyanobacteria; the functional role of most genes differentially expressed under a fluctuating P availability have not been characterized yet.

## Figures and Tables

**Figure 1 cells-09-01933-f001:**
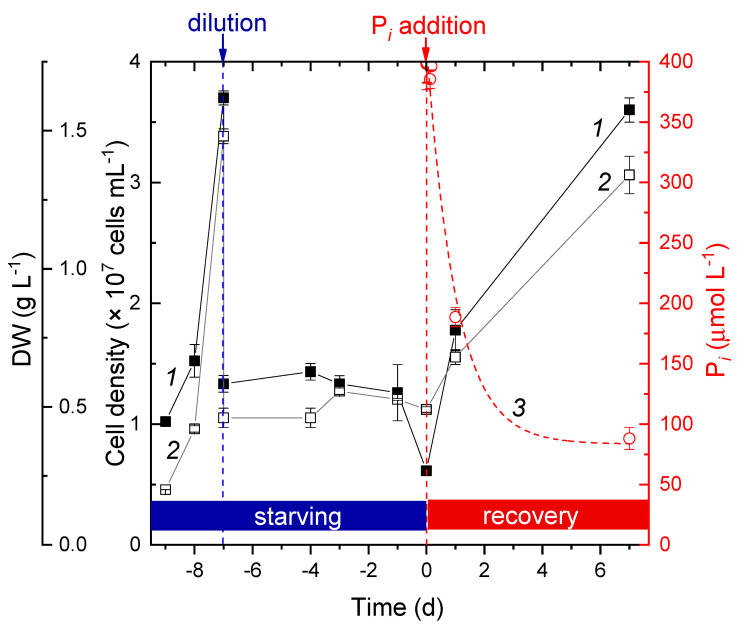
Changes in cell density (1) and cell dry weight (2) of the *Nostoc* sp. PCC 7118 during phosphorus starvation and after refeeding of the starved culture with inorganic phosphate (P*_i_*). The decline of the P*i* added to the medium is shown (3) (right scale). The moments of dilution of the culture and its refeeding with phosphorus are shown with arrows at the top.

**Figure 2 cells-09-01933-f002:**
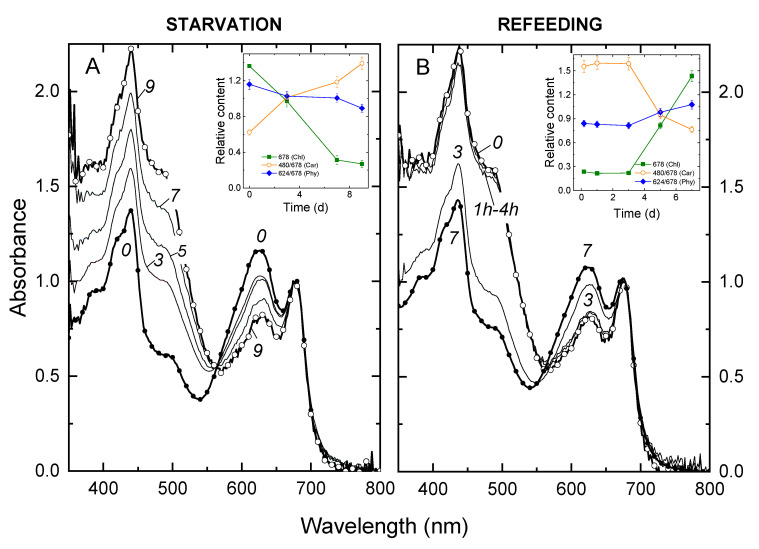
Changes in the absorption spectra of *Nostoc* sp. 7118 cultures during (**A**) phosphorous (P) starvation and (**B**) refeeding experiments. Each spectrum represents an average of six biological replicas normalized to the red chlorophyll absorption maximum (678 nm). Inserts: the changes of the absorption indices reflecting the contents of chlorophylls, carotenoids, and phycobilins in the culture plotted versus time. The time (d) after P deprivation (in **A**) or refeeding with Pi (in **B**) is shown near the corresponding curves or on the X-axes. In panel (**A**), the changes after dilution of the culture are shown (see Figure 1).

**Figure 3 cells-09-01933-f003:**
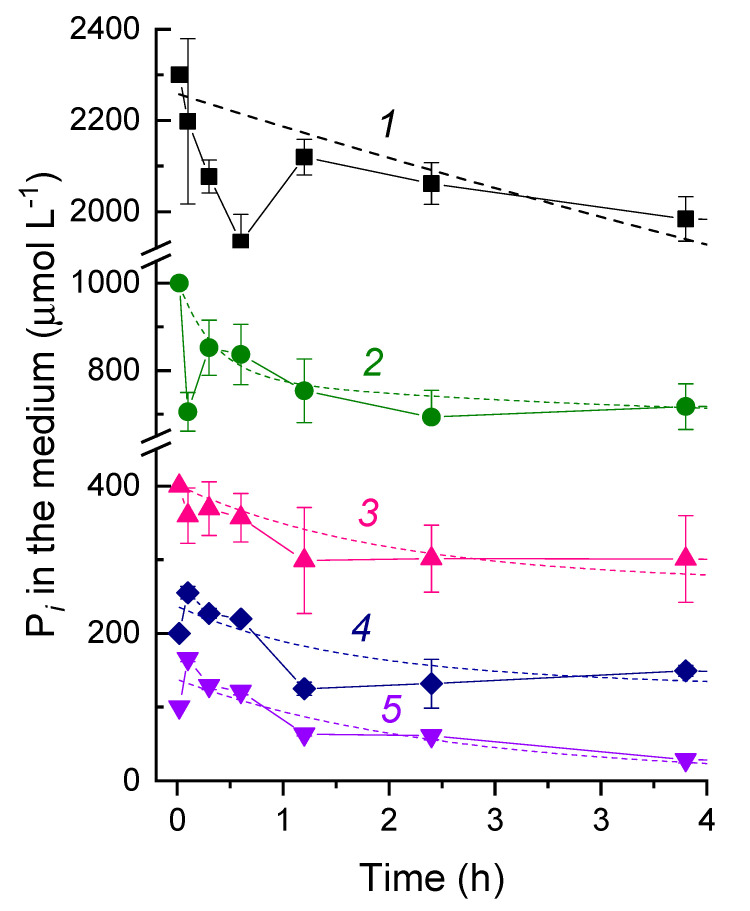
The kinetics of the changes in the external P*_i_* concentration after refeeding of the P-starved cells of *Nostoc* sp. PCC 7118 with different amounts of P*_i_* (final concentration, µmol L^−1^): 1–2300, 2–1000, 3–400, 4–200, and 5–100. The P*_i_* uptake was monitored during 24 h after refeeding; the data for the first 4 h are shown. Dashed line—the exponential best-fit function.

**Figure 4 cells-09-01933-f004:**
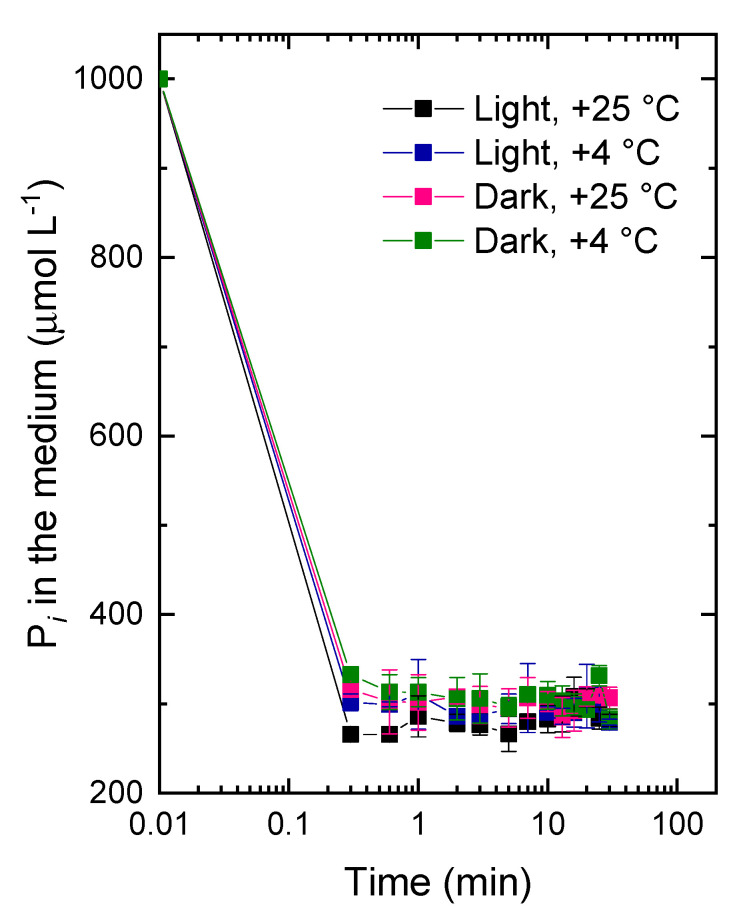
Effect of light and temperature on the fast phase of the uptake of P*_i_* added to the P-starved cells of *Nostoc* sp. PCC 7118. The values obtained after the 20th second of the experiment did not differ significantly.

**Figure 5 cells-09-01933-f005:**
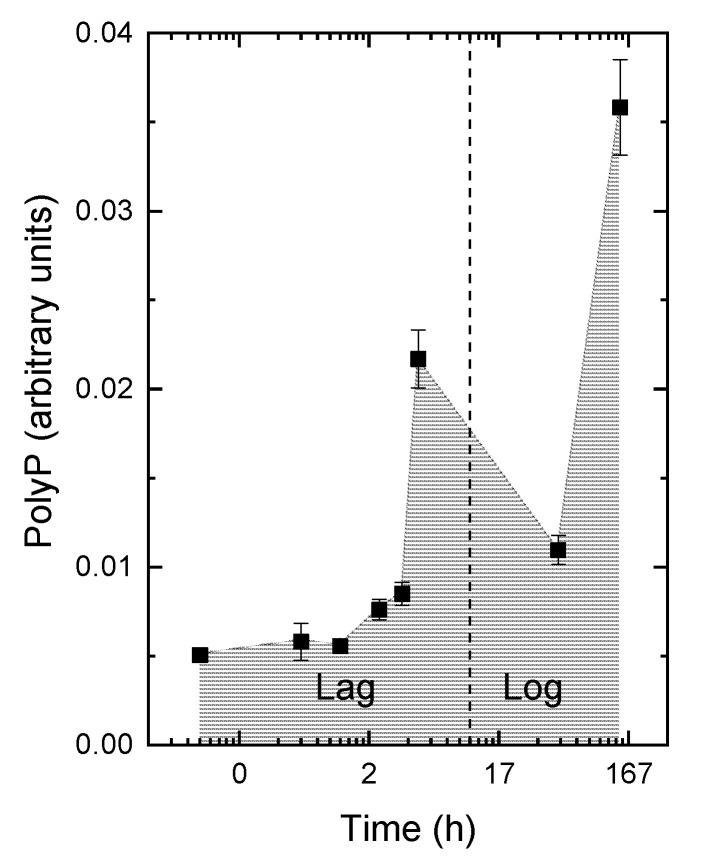
Changes of the cell polyphosphate content in *Nostoc* sp. PCC 7118 after refeeding of the P-starved cells with P*_i_*. Note the transient spike of cell polyphosphate taking place during the lag phase around the 4th h after refeeding. The polyphosphate (PolyP) are consumed during the log phase but reappear during the slow-down of cell division during the onset of the stationary phase. The units on the ordinate scale represent the background-corrected brightness of 4′,6-diamidino-2-phenylindole (DAPI)-stained PolyP in the cell (for details, see Supplementary methods and [39]).

**Figure 6 cells-09-01933-f006:**
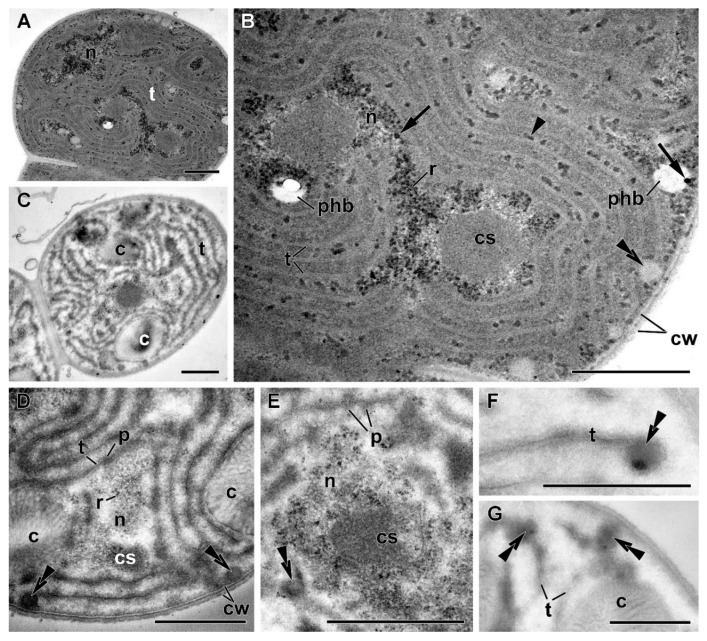
Ultrastructure of the cyanobacterium *Nostoc* sp. PCC 7118 (**A**,**B**) preculture grown in complete BG-11 medium or (**C**–**G**) in the P-free BG-11 (BG-11–P) medium. c—cyanophycin granules, cs—carboxysome, cw—cell wall, *n*—nucleoid, p—phycobilisomes, phb—granule of polyhydroxybutyrate, r—ribosomes, and t—thylakoid (s); arrowhead points to α-granule of glycogen, double arrowhead points to β-granule of the lipid, and arrow points to P-containing inclusion. Scale bars: 0.5 µm.

**Figure 7 cells-09-01933-f007:**
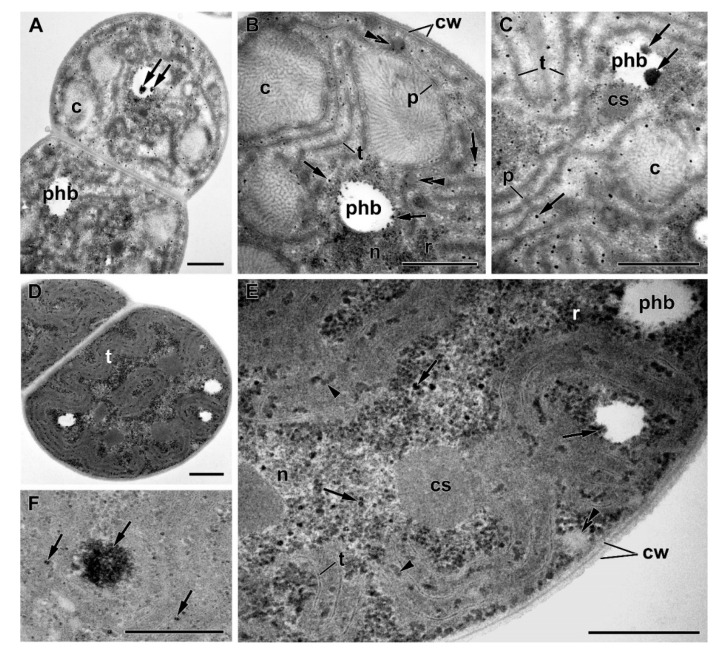
Ultrastructure of the cyanobacterium *Nostoc* sp. PCC 7118 cells (**A**–**C**) 4 h and (**D**–**E**) 7 d after refeeding of the P-starved cells with P*_i_*. c—cyanophycin granules, cs—carboxysome, cw—cell wall, *n*—nucleoid, p—phycobilisomes, phb—granule of polyhydroxybutyrate, r—ribosomes, and t—thylakoid(s); arrowhead points to α-granules of glycogen, double arrowhead points to β-granules of the lipid, and arrow points to a P-containing inclusion. Scale bars: 0.5 µm.

**Figure 8 cells-09-01933-f008:**
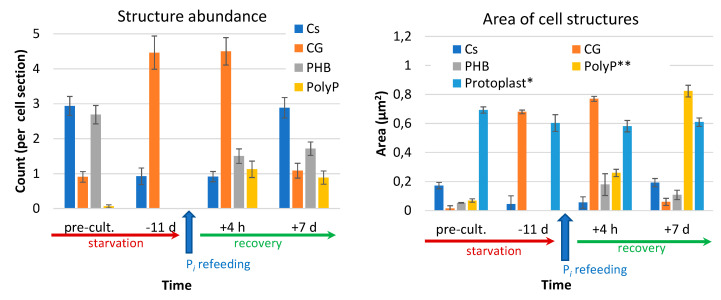
Changes in the abundance of the cell structures and inclusions (carboxysomes, Cs; cyanophycin granules, CG; polyhydroxybutyrate, PHB; and polyphosphate, PolyP) in the cells of *Nostoc* sp. PCC 7118 during P starvation and recovery from it. Average values ± SE are shown. * Protoplast area is shown as µ^2^ 0.1. ** PolyP grain area is shown as µ^2^ 100.

**Table 1 cells-09-01933-t001:** Changes in the expression level of the genes encoding ABC-type inorganic phosphate (P*_i_*) transporter/binding proteins involved in the response to the shortage of phosphorous (P) and, potentially, in its luxury uptake in *Nostoc* sp. PCC 7118. The tentative functional annotation was given according to the published genome of its parent strain *Nostoc* sp. PCC 7120 [23,33,49,52]. See also Appendix A.

ORF	Name (Description)	Conditions			
		Preculture (+P)	P-Starved (−P)	Recovery (+P)	
				1 d	7 d
all4573	*pstA1*	−1.92	−0.43	−0.47	−0.1
all4572	*pstB1*	−1.56	−0.28	−0.1	−0.58
all4574	*pstC1*	−2.76	−0.13	−0.29	−0.09
all4575	*pstS1*	−3.74	−0.18	−0.37	−0.69
all0909	*pstA2*	−2.76	0.88	−0.16	−1.92
all0908	*pstB2*	−0.94	−0.4	−0.9	−0.45
all0910	*pstC2*	−3.62	−0.37	−0.58	−0.79
all0911	*pstS2*	−6.09	0.95	−0.03	−0.78

**Table 2 cells-09-01933-t002:** Changes in the expression level of the genes involved in the P acquisition and polyphosphate metabolism in *Nostoc* sp. PCC 7118. The tentative functional annotation is given according to the published genome of its parent strain *Nostoc* sp. PCC 7120 [23,33,49,52].

ORF	Name (Description)	Conditions			
		Preculture (+P)	P-Starved (–P)	Recovery (+P)	
				1 d	7 d
alr2234	*phoD* (extracellular phosphatase)	−5.04	−0.4	−0.89	−2.55
all1686	*phoA* (alkaline phosphatase)	−0.16	0.91	0.36	0.01
all2843	alkaline phosphatase	−4.23	1.86	0.92	0.74
alr0276	*nucH* (extracellular nuclease)	−2.18	0.08	−0.17	0.31
all1371	*ppgK* (PolyP-dependent glucokinase)	0.38	−0.24	0.61	0.17
alr3593	*ppk1* (ATP-dependent PolyP kinase)	0.71	2.00	0.63	0.45
all2191	*ppk2* (GTP-dependent PolyP kinase)	−1.29	0.90	1.23	1.28
–	*ppa* (pyrophosphatase)	0.67	−0.99	0.83	−0.20
all3552	*ppx* (exopolyphosphatase)	0.06	−0.93	0.44	0.08

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
