# Peer review of "Phosphorus Feast and Famine in Cyanobacteria: Is Luxury Uptake of the Nutrient Just a Consequence of Acclimation to Its Shortage?"

_cells, 2020, doi:10.3390/cells9091933_

Round 1

Reviewer 1 Report

The manuscript provides original results on phosphorus uptake in Cyanobacteria, which could be a matter of interest of the journal readers. Experiments were well proposed, providing detailed information and the results are properly interpretted. The manuscript is written intelligibly and findings well supported by graphs. I have not found any objections to the form of manuscript as well as to its scientific value.

I would recommend the manuscript to be published as it is.

Author Response

We are grateful to the reviewer for his/her favorable evaluation of our manuscript.

Reviewer 2 Report

The paper is dedicated to investigation of acclimations of cyanobacteria to fluctuating phosphorus availability and dissecting the nature of luxury phosphorus uptake, the phenomenon scarcely explored so far. The authors present some interesting findings regarding the kinetics of inorganic phosphate uptake by the pre-starved cells of a non-diazotrophic cyanobacterium Nostoc sp. PCC 7118.

The strengths of the present study include using a broad array of modern methods including genomic and transcriptomic analysis as well as advanced analytical transmission electron microscopy yielding important mechanistic insights into the phenomena of nutrient uptake in cyanobacteria during abrupt transition from oligotrophic to eutrophic conditions.

The conclusion about the nature of luxury phosphorus uptake is non-trivial and inspiring for further studies in this field. This manuscript fits well into the scope of the Special Issue of Cells and can be accepted pending amendment of some issues/answering questions listed below.

- What could be possible effect of the cell envelope on the observed kinetics of Pi uptake? Some speculations are welcome.

- Could you suggest possible reasons for the observed changes in cyanophycin abundance (Fig. 8)?

- How the polyphosphate-containing cell inclusions were discerned on the cyanobacterial cell sections?

- Please introduce the FC = 1 line in Fig. S2 for better comprehension of the figure.

Additionally, literature survey needs improvement and please expand it. Additional related papers can be found in the Photosynthesis Res., Photosynthetica , i.e. many and very good papers has been published in these Journals on this subject. Please kindly add some of them to Refs list.

Next and most importantly, the language should be thoroughly revised and the presentation of the data – improved significantly.

Author Response

- What could be possible effect of the cell envelope on the observed kinetics of Pi uptake? Some speculations are welcome.

RESPONSE: we are thankful to the reviewer for driving indeed, some of the observed effects can be related to the barrier or transport function of the cell envelope which received scarce attention of researchers. In particular, some inorganic phosphate transporters were annotated in the genome of cyanobacteria but have never been functionally characterized. Some considerations are presented in the text of the manuscript (LL506-515).

- Could you suggest possible reasons for the observed changes in cyanophycin abundance (Fig. 8)?

RESPONSE: we believe that the cyanophycin can be the main depot for nitrogen liberated during decomposition of proteins as well as for excessive carbon photo-fixed during acclimation to P shortage (LL312-314).

- How the polyphosphate-containing cell inclusions were discerned on the cyanobacterial cell sections?

RESPONSE: we employed the method of analytical transmission electron microscopy which allows for discerning of P-containing cell inclusions by their characteristic emission and energy loss spectra (Fig. S1).

- Please introduce the FC = 1 line in Fig. S2 for better comprehension of the figure.

RESPONSE: done.

Additionally, literature survey needs improvement and please expand it. Additional related papers can be found in the Photosynthesis Res., Photosynthetica , i.e. many and very good papers has been published in these Journals on this subject. Please kindly add some of them to Refs list.

RESPONSE: according to the recommendation of the reviewer, we updated the reference list including latest reviews and some topical experimental papers, for example:

Collier, J. L., et al. (1994). "Changes in the cyanobacterial photosynthetic apparatus during acclimation to macronutrient deprivation." Photosynthesis Research 42(3): 173-183.

Goodenough, U., et al. (2019). "Acidocalcisomes: Ultrastructure, Biogenesis, and Distribution in Microbial Eukaryotes." Protist 170(3): 287-313.

Sanz-Luque, E., et al. (2020). "Polyphosphate: A Multifunctional Metabolite in Cyanobacteria and Algae." Frontiers in Plant Science 11.

de Siqueira Castro, J., et al. (2020). "Microalgae based biofertilizer: A life cycle approach." Science of The Total Environment.

Next and most importantly, the language should be thoroughly revised and the presentation of the data – improved significantly.

RESPONSE: we did our best to revise the English style and used this opportunity to correct typos in the text of the manuscript.

Reviewer 3 Report

The authors investigated cell structures and cell components as well as the expression of genes that are important for P metabolism under P-depleted conditions and after re-feeding of phosphate. They found differences in pigment composition, different cell constituents and gene expression. The strong expression of P uptake genes could be observed under starvation and for a certain time even after phosphate re-feeding. They conclude that this phenomenon is responsible for the luxury P uptake.

The manuscript is an important contribution to better understand the effects of P-deficiency on cyanobacteria as well as luxury P accumulation. Thus, this work  supports the findings of other authors, e.g., Nausch et al. (2009), who derived from P uptake experiments with radiotracers that P accumulation in cyanobacteria after P uptake can only be explained by upwelling if the supplied P is taken up at a rate similar to that prevailing under P-depleted conditions.

The authors have cited a large number of relevant literature. However, a new research should be carried out. In recent years new papers have been published, especially on Poly P.

Information on the Nostoc strain used is missing. Where was the strain obtained from and what is the basic organism for this strain (alternatively-how is the morphology). The genus Nostoc includes several species which are different in morphology and physiology, as well as pelagic and benthic species. This information is important to understand for which organism the results are representative.

The strain PCC7118 has lost not only heterocysts but also gas vacuoles due to the mutation. Otherwise, separation of the organisms from the medium by centrifugation would not be possible. Or is it a benthic organism?

The methods used (page 3 lines 95-137) are partly described very briefly  refering to literature. A detailed description could be provided as " supplemented material". This additional material should be prepared specifically for this manuscript and submitted together  with the manuscript. It is not  reasonable to search in the internet to understand the manuscript.

I am always fascinated when measurements  done in second-intervalls are presented. But I know that handling the samples lasted longer and it is difficult to get measurements in such short time intervals. Information on sample preparation for the individual parameters after sampling should be integrated in the manuscript.

The results section contains sentences that belong in the discussion. Examples are: lines 249-250, 256-260,275-278,312-314,356-360. There should be a clear separation between results and discussion.  In the results text, it could be partly more precise explained  what  the figures say. The results of gene expression should be supported by figures or tables, especially since the main message of the manuscript (slow deactivation of P uptake genes) and large parts of the discussion are based on these analyses.

The different time scales in the figures with the very large numbers are somewhat confusing. Perhaps seconds could be replaced by minutes (Fig.4) and minutes by hours (Fig.3 and 5). This would make it easier to compare the figures.

Discussion-Parts in the result section should be integrated into the chapter "discussion". The discussion should focus more strongly on the main result.

Specific comments

Page 1, line 16: „ We studied the LPU after re-feeding…“

Comment: „We studied the LPU before and after re-feeding… „

Page 1, line 24-25: „Our study revealed..“

Comment:  Cutting the sentence would inprove the understanding.

Page 2, line 52-53: Comment: Resulte of Braun et al. (2019)  could be included here.

Page 3, line 128: Comment: By P-species I mean PO4, DOP and POP (dissolved and particulate organic P). but here only phosphate and polyphosphate were measured. The title should be changed accordingly.

Page 5, Fig. 1 and other figures: „P (µM L-1)“

Comment: There are two ways of writing the unit: µM or µmolL-1.

The unit used here is wrong

Page 6, line 234 :“ The initial dip…“

Comment: What is dip? Correct the word ,please.

Page 7, line 246: „ To beter“

Comment: replace by“ to better“, please

Page 9, Fig.5 lable of axis PolyP (a.u.)

Comment, What is a.u.? It should be explained in the legend.

Page 9 ,line 288:“ electrpn-dense)

Comment:  Correct the word, please

Page 14 line 426: Comment: the words „under oligotrophic conditions“ should be replaced by „ under  phosphate depleted conditions“.

Author Response

The manuscript is an important contribution to better understand the effects of P-deficiency on cyanobacteria as well as luxury P accumulation. Thus, this work  supports the findings of other authors, e.g., Nausch et al. (2009), who derived from P uptake experiments with radiotracers that P accumulation in cyanobacteria after P uptake can only be explained by upwelling if the supplied P is taken up at a rate similar to that prevailing under P-depleted conditions.

RESPONSE: we are grateful to the reviewer for pointing us to this interesting work. It was added to the reference list.

The authors have cited a large number of relevant literature. However, a new research should be carried out. In recent years new papers have been published, especially on Poly P.

RESPONSE: we are thankful to the reviewer for his/her evaluation of our contribution. Indeed, the field of phosphorus and polyphosphate metabolism is very dynamic. According to the suggestion of the reviewer, we included the latest reviews, some of which appeared during the time of the revision:

Sanz-Luque, E., et al. (2020). "Polyphosphate: A Multifunctional Metabolite in Cyanobacteria and Algae." Frontiers in Plant Science 11.

Goodenough, U., et al. (2019). "Acidocalcisomes: Ultrastructure, Biogenesis, and Distribution in Microbial Eukaryotes." Protist 170(3): 287-313.

de Siqueira Castro, J., et al. (2020). "Microalgae based biofertilizer: A life cycle approach." Science of The Total Environment.

Information on the Nostoc strain used is missing. Where was the strain obtained from and what is the basic organism for this strain (alternatively-how is the morphology). The genus Nostoc includes several species which are different in morphology and physiology, as well as pelagic and benthic species. This information is important to understand for which organism the results are representative.

The strain PCC7118 has lost not only heterocysts but also gas vacuoles due to the mutation. Otherwise, separation of the organisms from the medium by centrifugation would not be possible. Or is it a benthic organism?

RESPONSE: The strain Nostoc sp. PCC 7118 (ATCC 27892) was obtained from the CALU collection in the beginning of the studies of cyanobacteria at the Biological Faculty of Lomonosov Moscow State University. It was tested side-by-side with the “parent” strain 7120 (obtained directly from PCC) and confirmed to be authentic (non-diazotrophic etc.). The molecular test of the taxonomic assignment (DNA barcoding) also confirmed the authenticity of the PCC 7118 we used (this was mentioned in the text of the manuscript). Therefore, we did not perform the cell segregation ourselves and postulate that the cultures used in the present study were authentic and free from contamination. According to the description on the PCC web site (https://catalogue-crbip.pasteur.fr/resultatRecherche.xhtml), “Gas vesicles: transient”. We normally do not observe gas vesicles in 7118 cells under our cultivation conditions. It is also stated in the description of the original strain on the collection web-site “Source unknown”, so it is difficult to say whether it is a benthic strain.

The methods used (page 3 lines 95-137) are partly described very briefly  refering to literature. A detailed description could be provided as " supplemented material". This additional material should be prepared specifically for this manuscript and submitted together  with the manuscript. It is not  reasonable to search in the internet to understand the manuscript.

RESPONSE: according to the recommendation of the reviewer, we provided a complete description of the mentioned methods in the supplementary material.

I am always fascinated when measurements  done in second-intervalls are presented. But I know that handling the samples lasted longer and it is difficult to get measurements in such short time intervals. Information on sample preparation for the individual parameters after sampling should be integrated in the manuscript.

RESPONSE: briefly, the first sample was taken ASAP after re-feeding and rapidly filtered through syringe filters to obtain the sample of the medium and immediately frozen in liquid nitrogen. These samples were stored at –20 °C before the assay of the residual Pi concentration. Therefore, actual sampling was done within several (18, 36, 60, 120, 180, 300, 420, 600, 780, and 960) seconds, but the analyses were done later, at a longer timescale. Corresponding clarifications were added to the section 2.2. of the manuscript.

The results section contains sentences that belong in the discussion. Examples are: lines 249-250, 256-260,275-278,312-314,356-360. There should be a clear separation between results and discussion.  In the results text, it could be partly more precise explained what  the figures say.

RESPONSE: following the recommendation of the reviewer, we moved the mentioned passages from the Results to the Discussion section and refined the description of the figures in the Results section.

The results of gene expression should be supported by figures or tables, especially since the main message of the manuscript (slow deactivation of P uptake genes) and large parts of the discussion are based on these analyses.

RESPONSE: the complete results of the differential expression analysis are too extensive to keep in the main body of the paper, so they are placed to the supplementary materials. Nevertheless, in accord with the suggestion of the reviewer, we moved the most important data to the main body of the revised manuscript (table 1 and table 2).

The different time scales in the figures with the very large numbers are somewhat confusing. Perhaps seconds could be replaced by minutes (Fig.4) and minutes by hours (Fig.3 and 5). This would make it easier to compare the figures.

RESPONSE: we converted the timescale of Fig. 4 to make uniform time units for X axes in figs. 3–5. We attempted to convert hours to minutes in other figures, but the resulting scale was too coarse and actually complicated the understanding of the figures.

Discussion-Parts in the result section should be integrated into the chapter "discussion". The discussion should focus more strongly on the main result.

RESPONSE: please see our reply above.

Specific comments

Page 1, line 16: „ We studied the LPU after re-feeding…“

Comment: „We studied the LPU before and after re-feeding… „

RESPONSE: we would like to keep the current wording because there was no Pi in the medium and hence no LPU before the re-feeding.

Page 1, line 24-25: „Our study revealed..“

Comment:  Cutting the sentence would inprove the understanding.

RESPONSE: We split the following sentence into two shorter sentences to make this text more comprehensible.

Page 2, line 52-53: Comment: Resulte of Braun et al. (2019)  could be included here.

RESPONSE: we would gladly include the reference suggested by the reviewer to the literature review, however literature search:

http://apps.webofknowledge.com/Search.do?product=UA&SID=C33S57iJXFFafAZcXb3&search_mode=GeneralSearch&prID=49de29d3-9001-49a3-8650-29a8d239e110

revealed 43 papers but none relevant to the field of the present study.

Page 3, line 128: Comment: By P-species I mean PO4, DOP and POP (dissolved and particulate organic P). but here only phosphate and polyphosphate were measured. The title should be changed accordingly.

RESPONSE: we agree. The title of the subsection 2.5 was changed to “Assay of inorganic phosphate in the medium and polyphosphate in the cells”.

Page 5, Fig. 1 and other figures: „P (µM L-1)“

Comment: There are two ways of writing the unit: µM or µmolL-1.

The unit used here is wrong

RESPONSE: the units have been changed to µmol L-1 in the figures’ axes annotations and in the text. 

Page 6, line 234 :“ The initial dip…“

Comment: What is dip? Correct the word ,please.

RESPONSE: replaced with “decline”. 

Page 7, line 246: „ To beter“

Comment: replace by“ to better“, please

RESPONSE: done, thank you. 

Page 9, Fig.5 lable of axis PolyP (a.u.)

Comment, What is a.u.? It should be explained in the legend.

RESPONSE: it is ‘arbitrary units.’ We expanded the abbreviation in the annotation to figure’s Y axis.

Page 9 ,line 288:“ electrpn-dense)

Comment:  Correct the word, please

RESPONSE: done, thank you. 

Page 14 line 426: Comment: the words „under oligotrophic conditions“ should be replaced by „ under  phosphate depleted conditions“.

RESPONSE: corrected.

Reviewer 4 Report

Review of Cells article “Phosphorus feast and famine….” by Solovchenke et al. 2020

This paper attempts to determine the time response of luxury phosphate accumulation in a cyanobacterial strain (Nostoc sp.) following the addition of a very high concentration of  orthophosphate to P-starved cells. Unfortunately, the experiments were poorly designed and changes in cellular P concentration were never reported. Pi uptake should have been determined from the change in cellular P with time, expressed ideally as P per unit of cell biomass or cell carbon. However, this was never done. The authors could have computed the cellular P taken up per cell (since they measured cell concentrations) and the decrease in Pi concentration in the medium due to cellular uptake, but these calculations were never made or much less presented. And even if they were Pi limitation typically causes an increase in cell size so presenting cellular concentrations as P per cell can be misleading because of changes in cell size. Furthermore, changes in cellular P concentrations are determined by the relative balance between increases owing to P uptake and decreases owing to biodilution by cellular grow (i.e. production of new cell biomass). Thus, an increase in cellular P concentration can be caused by an increase in P uptake rate or a decrease in the specific cellular growth rate. Unfortunately, growth rates were never reported, nor were the cultures monitored frequently enough to discern the specific growth rate at the time samples were taken. Thus, there was no way to tell if the cells were growing exponentially and rapidly or if they had depleted one or more resources causing their growth rate to slow and eventually stop. The authors allude to the appearance of a second peak in polyphosphate being associated with the onset of stationary phase (a slowing and eventual cessation of growth), but present no data to document this. Finally, the presentation of the cellular polyphosphate data is inadequate. The data is presented as PolyP (a.u.) but there is no mention of what a.u. stands for either in the legend or the methods section. A.u. might mean arbitrary units, which is not very informative. And even if the units were relative PolyP per cell, the data would be influenced by P-induced changes in cell size, which were never examined.

Specific comments

Ln 91 – The Chl a concentration should be given in molar units, i.e., mmol/L.

Ln 109 – To measure P uptake it would have been better to directly measure the P content of the cells.

Ln 201 – The cell division rate not only “declined” with P starvation it stopped. This should be noted.

Ln 202 – The authors need to define “DW” here.

Lns 203&204 – The statement that the cell rapidly resumed cell division after Pi reintroduction at a “rate close to that of the P-sufficient pre-cultures” is not necessarily born out by the data plotted in Fig. 1. To show this, the cell density data should be plotted on a log scale since the specific cell division rate equals the change in the natural log of cell density per unit time. Once this is done equal cell division rate would be indicated by equal slopes of the time-dependent plots of log cell density. Also it would be useful to compute the average specific cell division rate after Pi addition to the P-starved cells and compare this value to the maximum growth rate observed under P-sufficiency.

Lns 204-206 – The onset of light limitation in the dense cultures complicates the interpretation of P-recovery in the cultures.

Ln 258 – The phrase “Pi taken up by the cell during is converted to PolyP.” does not make sense and needs revision.

Figure 5 - The units for cellular PoloyP need to be given in the legend. It’s not clear what a.u. refers to. The units should be moles Poly-P per cell.

Lns 266-268 – What is the reason for the second stationary phase onset at ~7 days. Usually it occurs when some critical nutrient becomes depleted, but that nutrient cannot be Pi because as is shown in Fig. 1, there is still ample Pi in the medium at that time to support culture growth. It would have been useful if more data time points were recorded in the recovery culture.

Lns 266-268 - The second peak in PolyP with the onset of stationary phase in the P-recovery culture likely was caused the decrease in cellular growth and division. Cellular P concentrations are determined by the balance between uptake and dilution by cell growth. Once the cells stopped growing, the decrease in growth rate would have caused the cellular Pi to increase, which would have supplied higher Pi levels to promote PolyP formation.

Fig. 6 – Were the cells shown in this figure growing exponentially? This is an important detail for interpretation of the micrographs.

Lns 283&284 – The meaning of this sentence is unclear. Some words appear to be missing.

Ln 285 – Revise wording to “….carboxysomes, which contain RuBisCo.”

Ln 323 – What was the growth rate of the cells on the 7th day after refeeding? Healthy P-sufficient cells with healthy photosynthetic machinery should have been growing at or near their maximum rate. Unfortunately, this cannot be determined from the sparse amount of cell data presented in Fig. 1. Earlier, the authors had stated that the cells had reached stationary phase by day 7.

Ln 398 – Meaning here is unclear. A word seems to be missing. Shouldn’t it be that “genes that control phosphonate transport………..were upregulated under phosphorus starvation…”. The word starvation is missing.

Ln 426&427 – Where is the evidence for this statement. There needs to be a citation. A safer statement would be that “Many photosynthetic microorganisms including many cyanobacteria evolved in nutrient-poor environments or environments with varying nutrient concentrations.”

Ln 428 – Revise wording to “…including the capability for luxury uptake.”

Ln 433-436 – The meaning of this sentence is unclear. Please revise.

Ln 499-502 – Cells typically have both high and low affinity transport systems. High affinity transport systems typically transport Pi at low external Pi concentrations and invariably are energy dependent and upregulated at low external Pi levels. Low affinity systems operate at high Pi levels and may or may not be energy dependent, have high Vmax values and may also be constitutive. Given the fast uptake rate and apparent lack of energy dependence at the high added level of Pi, it is likely that the fast initial uptake of Pi was not through an upregulated high affinity system, but rather through a low-affinity system.

Ln 516 – Typo. Should be “..transfer a large amount of Pi….”.

Ln 523 – The decrease in polyP following its cellular accumulation appears to coincide with the resumption of cellular growth and cell division. If so the temporary decrease observed may reflect the cellular consumption of polyP needed for cellular growth. Note that polyP not only supplies intracellular Pi needed for growth and reproduction, but also cellular energy in the form of ATP.

Lns 565-568 – There is no evidence that the high Pi uptake following Pi addition is from high affinity Pi transporters. The apparent lack of energy dependence for the uptake suggests that it’s not.

Stopped at 562

Author Response

We are deeply thankful to the reviewer for his/her constructive and thoughtful analysis of our manuscript and pointing us to its shortcomings. Please find below our item-by-item responses to the issues raised in the reviewer’s comments.

IMPORTANT: please see the attached file for inserted graphs.

This paper attempts to determine the time response of luxury phosphate accumulation in a cyanobacterial strain (Nostoc sp.) following the addition of a very high concentration of  orthophosphate to P-starved cells. Unfortunately, the experiments were poorly designed and changes in cellular P concentration were never reported. Pi uptake should have been determined from the change in cellular P with time, expressed ideally as P per unit of cell biomass or cell carbon. However, this was never done.

RESPONSE: we share the concern of the reviewer, so we would like to note that the measurements of P cell content were actually done, although not routinely. At certain points of the experiments, we chemically assayed total cell P content, in addition to the assay of the residual Pi in the medium. The analyses showed that the accumulation of P in the cells corresponded, with a reasonable precision (±10%), to the depletion of Pi in the medium under our experimental conditions. Since the analysis of residual Pi in the medium is much simpler, we decided to employ it routinely (see also our response to your comment to Ln 91 below). Expression of P per unit C might not be the best normalization in this particular case since the C content of the cells changes dramatically during P starvation and subsequent P replenishment. We added corresponding clarifications and a description of the cell P content assay it the revised manuscript.

The authors could have computed the cellular P taken up per cell (since they measured cell concentrations) and the decrease in Pi concentration in the medium due to cellular uptake, but these calculations were never made or much less presented.

RESPONSE: it was done, and the results of the calculations were incorporated into the results section of the revised manuscript. By the 1st day after re-feeding (during the fast phase of Pi uptake), the culture took up 18 pmol Pi/cell; during the next 6 days the uptake rate slowed down comprising 3.7 pmol Pi/cell. Corresponding values were incorporated into the text of the revised manuscript.

And even if they were Pi limitation typically causes an increase in cell size so presenting cellular concentrations as P per cell can be misleading because of changes in cell size.

RESPONSE: under our experimental conditions, Pi limitation did not lead to a dramatic increase in cell size as can be judged from the electron microscopy data (Figs. 6-8). However, there was an increase in cell weight (see the graph below) during P starvation. This increase in cell weight was mostly reversed after the Pi re-feeding.

Furthermore, changes in cellular P concentrations are determined by the relative balance between increases owing to P uptake and decreases owing to biodilution by cellular grow (i.e. production of new cell biomass). Thus, an increase in cellular P concentration can be caused by an increase in P uptake rate or a decrease in the specific cellular growth rate.

RESPONSE: the reasoning of the reviewer is perfectly valid, however we would like to draw attention to the fact that the bulk of the Pi added to the P-starved culture is normally consumed during the lag phase (first several hours after re-feeding, figs. 3 and 4). During that phase (which was actually in the focus of this work), cell division did not yet resume (µ ~ 0) so no significant biodilution is expected. Later, both processes (Pi uptake and biomass accumulation) continued side-by-side. Accordingly, the highest P percentages of DW (4.2%) were recorded during the initial (fast) phase of Pi uptake. Later it declined to 1.2–2% typical of P-sufficient phototroph cells.

Unfortunately, growth rates were never reported, nor were the cultures monitored frequently enough to discern the specific growth rate at the time samples were taken.Thus, there was no way to tell if the cells were growing exponentially and rapidly or if they had depleted one or more resources causing their growth rate to slow and eventually stop.

RESPONSE: we cannot agree completely with this note of the reviewer. The key nutrients were added (replenished) to the medium in known excess. The availability of N and P was monitored throughout the experiments and it was established that none of the nutrients was depleted after re-feeding. At the P-starvation stage, only P was absent in the medium and the culture was diluted with the fresh BG11–P medium ensuring that there was no light limitation as well. Please see also our reply to the specific comments to Lns 203&204 below.

The authors allude to the appearance of a second peak in polyphosphate being associated with the onset of stationary phase (a slowing and eventual cessation of growth), but present no data to document this.

RESPONSE: please take a note of the data showed in Fig. 5. Please see also our reply to the next comment and our reply to the specific comment to Fig. 5 below.

Finally, the presentation of the cellular polyphosphate data is inadequate. The data is presented as PolyP (a.u.) but there is no mention of what a.u. stands for either in the legend or the methods section. A.u. might mean arbitrary units, which is not very informative. And even if the units were relative PolyP per cell, the data would be influenced by P-induced changes in cell size, which were never examined.

RESPONSE: we apologize for the confusion. The units represent the brightness of fluorescence emission of PolyP stained with DAPI which is a standard assay for PolyP in situ (please see the references below). We agree that this kind of assay has its limitations since it reveals only relative changes in PolyP abundance, but it can be sufficient for the purpose of the present study (one needs to see significant peaks of PolyP accumulation). In the revised version, we specified in detail the PolyP assay type in the supplementary methods as well as in the legend to Fig. 5. Considering the changes in cell size, please see the Fig. 8 (the bars designated “protoplast”: the cell size did not change significantly throughout the experiment). Please see also our reply to the next comment and our reply to the specific comment to Fig. 5 below.

Voronkov, A. and M. Sinetova (2019). "Polyphosphate accumulation dynamics in a population of Synechocystis sp. PCC 6803 cells under phosphate overplus." Protoplasma.

Pokhrel, A., et al. (2019). "Assaying for Inorganic Polyphosphate in Bacteria." J Vis Exp(143).

MoudÅ™íková, Š., et al. (2017). "Quantification of polyphosphate in microalgae by raman microscopy and by a reference enzymatic assay." Analytical chemistry 89(22): 12006-12013.

Specific comments

Ln 91 – The Chl a concentration should be given in molar units, i.e., mmol/L.

RESPONSE: a quick literature analysis shows that mass units are also plausible and accepted, but the molar concentration has been added to the text of the revised manuscript.

Ln 109 – To measure P uptake it would have been better to directly measure the P content of the cells.

RESPONSE: although the only reason for a decline of the residual Pi in the medium, within the physiological pH range, is its uptake by the cells, we agree with the reviewer that direct measurements of internal P can be informative. Although it was not done routinely, at certain point of the experiments we chemically assayed total cell P content, in addition to the assay of the residual Pi in the medium. The results of the analyses showed that the accumulation of P in the cells coincides reasonably with the depletion of Pi in the medium under our experimental conditions. Since the latter analysis is much simpler technically, we decided to employ it routinely. We added corresponding clarifications and description of the cell P content assay it the revised manuscript.

Ln 201 – The cell division rate not only “declined” with P starvation it stopped. This should be noted.

RESPONSE: yes, indeed. The cessation of cell division has been used as reliable criterion of P starvation. A corresponding note was added to the text of the revised version.

Ln 202 – The authors need to define “DW” here.

RESPONSE: it was defined above (section 2.1).

Lns 203&204 – The statement that the cell rapidly resumed cell division after Pi reintroduction at a “rate close to that of the P-sufficient pre-cultures” is not necessarily born out by the data plotted in Fig. 1. To show this, the cell density data should be plotted on a log scale since the specific cell division rate equals the change in the natural log of cell density per unit time. Once this is done equal cell division rate would be indicated by equal slopes of the time-dependent plots of log cell density. Also it would be useful to compute the average specific cell division rate after Pi addition to the P-starved cells and compare this value to the maximum growth rate observed under P-sufficiency.

RESPONSE: We eliminated the text “rate close to that of the P-sufficient pre-cultures” from the revised text of the manuscript since it, in our opinion, is not at the core message of the study. However, the cell density graph shown in Fig. 1 but in log scale is presented below:

The magenta dashed lines show that the slopes of the cell density change over the periods in question were close as the corresponding computed µ values (0.35 vs. 0.46). Since we added the changes of cell DW to the same graph in the revised version of the manuscript, we would like to keep the original graph in linear scale (but we added the µ values to the revised manuscript). Please see also our reply to the general comments of the reviewer above.

Lns 204-206 – The onset of light limitation in the dense cultures complicates the interpretation of P-recovery in the cultures.

RESPONSE: light limitation is expectable to build up in batch cultures when other resources are ample as in our case after Pi re-feeding. On one hand, it is an added complexity in comparison with e.g. continuous cultures. On the other hand, it is essential for the experiment since one would hardly see accumulation of PolyP without a slowdown of cell division (due to e.g. light limitation) on the background of sufficient Pi presence in the medium.

Ln 258 – The phrase “Pi taken up by the cell during is converted to PolyP.” does not make sense and needs revision.

RESPONSE: corrected. Thank you for pointing us to this.

Figure 5 - The units for cellular PoloyP need to be given in the legend. It’s not clear what a.u. refers to. The units should be moles Poly-P per cell.

RESPONSE: the units represent the brightness of fluorescence emission of PolyP stained with DAPI which is a standard assay for PolyP in situ (please see the references below). We agree that this kind of assay has its limitations since it reveals only relative changes in PolyP abundance, but it can be sufficient for the purpose of the present study (one needs to see significant peaks of PolyP accumulation). In the revised version, we specified in detail the PolyP assay in the supplementary methods as well as in the legend to Fig. 5.

Voronkov, A. and M. Sinetova (2019). "Polyphosphate accumulation dynamics in a population of Synechocystis sp. PCC 6803 cells under phosphate overplus." Protoplasma.

Pokhrel, A., et al. (2019). "Assaying for Inorganic Polyphosphate in Bacteria." J Vis Exp(143).

MoudÅ™íková, Š., et al. (2017). "Quantification of polyphosphate in microalgae by raman microscopy and by a reference enzymatic assay." Analytical chemistry 89(22): 12006-12013.

Lns 266-268 – What is the reason for the second stationary phase onset at ~7 days. Usually it occurs when some critical nutrient becomes depleted, but that nutrient cannot be Pi because as is shown in Fig. 1, there is still ample Pi in the medium at that time to support culture growth. It would have been useful if more data time points were recorded in the recovery culture.

RESPONSE:  the reviewer correctly notes that the stationary phase onset cannot be due to a critical nutrient limitation. We further confirmed routinely (by ion-exchange HPLC and other techniques, please see the Methods) that there was also plenty of inorganic N remaining in the medium. The observed onset of stationary phase by the 7th d of the experiment was due to light limitation. This assumption is supported by the data on optical density of the culture which was not diluted after Pi re-feeding (Fig. 2) and by an increase of the quantum yield of photosystem 2 (Qy, Fv/Fm) measured by a PAM fluorometer (the latter data were not included in the manuscript for the sake of brevity). Normally, a nutrient limitation leads to a decline in Fv/Fm instead of its increase due to non-photochemical quenching buildup.

Lns 266-268 - The second peak in PolyP with the onset of stationary phase in the P-recovery culture likely was caused the decrease in cellular growth and division. Cellular P concentrations are determined by the balance between uptake and dilution by cell growth. Once the cells stopped growing, the decrease in growth rate would have caused the cellular Pi to increase, which would have supplied higher Pi levels to promote PolyP formation.

RESPONSE: this is exactly the point we tried to make, nicely formulated by the reviewer!

Fig. 6 – Were the cells shown in this figure growing exponentially? This is an important detail for interpretation of the micrographs.

RESPONSE: we do agree. In the panels A and B, the cells were sampled from the exponentially growing pre-cultures (“The pre-cultures were maintained at the exponential phase by daily dilution with the same medium.”, please see the methods) are shown. In the panels C–G, the cells from P-limited cultures are shown. The legend to figure 6 has been amended.

Lns 283&284 – The meaning of this sentence is unclear. Some words appear to be missing.

RESPONSE: it was corrected. Thank you.

Ln 285 – Revise wording to “….carboxysomes, which contain RuBisCo.”

RESPONSE: done.

Ln 323 – What was the growth rate of the cells on the 7th day after refeeding? Healthy P-sufficient cells with healthy photosynthetic machinery should have been growing at or near their maximum rate. Unfortunately, this cannot be determined from the sparse amount of cell data presented in Fig. 1. Earlier, the authors had stated that the cells had reached stationary phase by day 7.

RESPONSE: the average specific growth rate over days 2-6 was 0.05 (mentioned in the revised version of the manuscript). Yes, the culture did so (grew at the maximum rate observed in our experiments, µ = 0.46). As a result, self-shading took place (it was a batch culture which was not diluted after the Pi re-feeding; please see also our reply to the general comments above) resulting in the onset of stationary phase by the 7th d after the Pi re-feeding. Please see our reply to your comments regarding Lns 203&204, Lns 266-268 above.

Ln 398 – Meaning here is unclear. A word seems to be missing. Shouldn’t it be that “genes that control phosphonate transport………..were upregulated under phosphorus starvation…”. The word starvation is missing.

RESPONSE: corrected. Sorry for this and thanks to you.

Ln 426&427 – Where is the evidence for this statement. There needs to be a citation. A safer statement would be that “Many photosynthetic microorganisms including many cyanobacteria evolved in nutrient-poor environments or environments with varying nutrient concentrations.”

RESPONSE: agreeing with the reviewer, we followed his/her recommendations.

Ln 428 – Revise wording to “…including the capability for luxury uptake.”

RESPONSE: done.

Ln 433-436 – The meaning of this sentence is unclear. Please revise.

RESPONSE: done.

Ln 499-502 – Cells typically have both high and low affinity transport systems. High affinity transport systems typically transport Pi at low external Pi concentrations and invariably are energy dependent and upregulated at low external Pi levels. Low affinity systems operate at high Pi levels and may or may not be energy dependent, have high Vmax values and may also be constitutive. Given the fast uptake rate and apparent lack of energy dependence at the high added level of Pi, it is likely that the fast initial uptake of Pi was not through an upregulated high affinity system, but rather through a low-affinity system.

RESPONSE: we appreciate the thoughtful comment of the reviewer and paid it a close attention below:

>> High affinity transport systems typically transport Pi at low external Pi concentrations and invariably are energy dependent and upregulated at low external Pi levels.

True. We see this in the cells acclimated low or no Pi in the medium (see e.g. table 1 in the revised manuscript).

>> Low affinity systems operate at high Pi levels and may or may not be energy dependent, have high Vmax values and may also be constitutive.

To the best of our knowledge, low-affinity Pi transport system has not yet been characterized in cyanobacteria and, in particular, in the studied Nostoc. Our genome mining effort indicated that the genomes of PCC 7118 and PCC 7120 lack genes resembling the low-affinity Pi transporters from Pht- or Pho-family typical for heterotroph bacteria or eukaryotes. Some hits were found (but with a poor identity of 33%) for the PitA transporter (alr3096, alr2336). These genes did not show a remarkable differential expression pattern under alternating Pi availability so the contribution of these transporters, if any, is likely constitutive.

Considering that low-affinity Pi transporter system might be dispensable for growth of microorganisms (Gebhard, S. et al. 2009. The low-affinity phosphate transporter PitA is dispensable for in vitro growth of Mycobacterium smegmatis. BMC microbiology, 9(1)), the existence and operation of such transporter system in the Nostoc studied in our work is questionable.

>> Given the fast uptake rate and apparent lack of energy dependence at the high added level of Pi, it is likely that the fast initial uptake of Pi was not through an upregulated high affinity system, but rather through a low-affinity system.

This is an important consideration. Indeed, one cannot rule out a certain contribution of a constitutive low-affinity Pi transport to the LPU phenomenon (though it remains so far elusive—please see above). But the data obtained do not support the major contribution of these transporters since we do not see a massive Pi uptake upon pre-suspending of the cells acclimated to ample Pi in a Pi-enriched medium. Collectively, we see the dramatic (i) Pi uptake and (ii) transient PolyP accumulation only in the cells with up-regulated high-affinity Pi transporters. Still some constitutive uptake via a low-affinity transporter(s) can take place since we later see a buildup of PolyP during a slowdown of cell division. Moreover, there could be still unknown Pi transporters hiding in the uncharacterized part of the Nostoc genome…

We enriched the discussion in the revised version of the manuscript with the points made above.

Ln 516 – Typo. Should be “..transfer a large amount of Pi….”.

RESPONSE: corrected.

Ln 523 – The decrease in polyP following its cellular accumulation appears to coincide with the resumption of cellular growth and cell division. If so the temporary decrease observed may reflect the cellular consumption of polyP needed for cellular growth. Note that polyP not only supplies intracellular Pi needed for growth and reproduction, but also cellular energy in the form of ATP.

RESPONSE: this is another important point we tried to make, also grasped perfectly by the reviewer! We mentioned the energy storing function of PolyP in the introduction.

Lns 565-568 – There is no evidence that the high Pi uptake following Pi addition is from high affinity Pi transporters. The apparent lack of energy dependence for the uptake suggests that it’s not.

RESPONSE: yes, it would be so under conditions when high-affinity Pi transporters normally operate (= low Pi concentration in the medium). However, an abrupt transition from zero external Pi to a very high (relative to natural conditions) external Pi leads to the situation when the high-affinity Pi transporters can just allow the Pi to flow inside the cell without the need of additional energy input. It might be the case judging from thermodynamic calculations (Ritchie, R. J., et al. 2001. "Phosphate limited cultures of the cyanobacterium Synechococcus are capable of very rapid, opportunistic uptake of phosphate." New Phytologist 152(2): 189-201, cited in the manuscript) and data presented in Fig. 3. Of course, it is a hypothesis which needs further mechanistic proof. Please see also our reply to the specific comments to Ln 499-502 above.

Round 2

Reviewer 3 Report

The authors have revised the manuscript taking into account most of the comments. It is now more consistent and easier to understand. There are still some minor errors that need to be corrected before publication.

Line 95: what does the number 0.034 mmol L-1 refer to? Chlorophyll is not given in mmol L-1. The conversion to Kohlenstoff is vague, because the conversion factor can be very different

Line 148: Phosphate is an inorganic compound. The word " inorganic" can be deleted

Line 155: Do you mean 2.7 mmol L-1and 0.3 mmol L-1? Correct it, please.

Line 455: „…polyphosphate metabolism in Nostoc sp. PCC 7118. The tentative functional annotation ois ….“

Figures 1,3 and 4: In these figures, the units are corrected in the upper pictures , but it is not done  in the figures below. Please correct the units here.

Author Response

Line 95: what does the number 0.034 mmol L refer to? Chlorophyll is not given in mmol L .

RESPONSE: we excluded the molar amount and left only mass amount in text of the revised version.

The conversion to Kohlenstoff is vague, because the conversion factor can be very different

RESPONSE: we did not measure the actual amount of dissolved inorganic carbon in the medium assuming that it was supplied in excess as 2% CO2 (by volume) mix with the air for bubbling the cultures and to keep pH in the optimal range (7..8). Corresponding not was added to the text in the Methods section.

Line 148: Phosphate is an inorganic compound. The word " inorganic" can be deleted

RESPONSE: done.

Line 155: Do you mean 2.7 mmol L and 0.3 mmol L ? Correct it, please.

RESPONSE: done.

Line 455: „…polyphosphate metabolism in Nostoc sp. PCC 7118. The tentative functional annotation ois ….“

RESPONSE: corrected.

Figures 1,3 and 4: In these figures, the units are corrected in the upper pictures , but it is not done  in the figures below. Please correct the units here.

RESPONSE: done.